# An Image is Worth 32 Tokens for Reconstruction and Generation

**Qihang Yu[1*], Mark Weber[1,2*], Xueqing Deng[1], Xiaohui Shen[1], Daniel Cremers[2], Liang-Chieh Chen[1]**

[1] ByteDance      [2] Technical University Munich      * equal contribution

https://yucornetto.github.io/projects/titok.html

*32 tokens can work well for...*

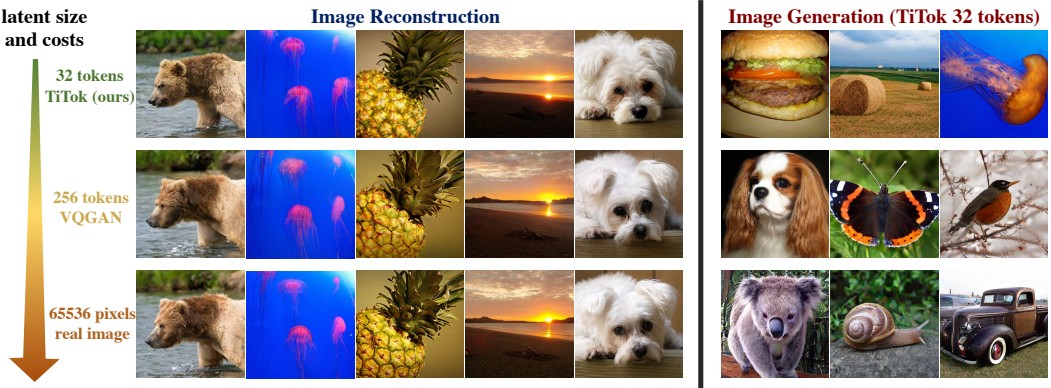

Figure 1: We propose **TiTok**, a compact **1D** tokenizer leveraging region redundancy to represent an image with only **32** tokens for image reconstruction and generation.

## Abstract

Recent advancements in generative models have highlighted the crucial role of image tokenization in the efficient synthesis of high-resolution images. Tokenization, which transforms images into latent representations, reduces computational demands compared to directly processing pixels and enhances the effectiveness and efficiency of the generation process. Prior methods, such as VQGAN, typically utilize 2D latent grids with fixed downsampling factors. However, these 2D tokenizations face challenges in managing the inherent redundancies present in images, where adjacent regions frequently display similarities. To overcome this issue, we introduce **T**ransformer-based 1-**D**imensional **Tok**enizer (TiTok), an innovative approach that tokenizes images into 1D latent sequences. TiTok provides a more compact latent representation, yielding substantially more efficient and effective representations than conventional techniques. For example, a $256 \times 256 \times 3$ image can be reduced to just **32** discrete tokens, a significant reduction from the 256 or 1024 tokens obtained by prior methods. Despite its compact nature, TiTok achieves competitive performance to state-of-the-art approaches. Specifically, using the same generator framework, TiTok attains **1.97** gFID, outperforming MaskGIT baseline significantly by 4.21 at ImageNet $256 \times 256$ benchmark. The advantages of TiTok become even more significant when it comes to higher resolution. At ImageNet $512 \times 512$ benchmark, TiTok not only outperforms state-of-the-art diffusion model DiT-XL/2 (gFID 2.74 *vs*. 3.04), but also reduces the image tokens by **64×**, leading to **410×** **faster** generation process. Our best-performing variant can significantly surpass DiT-XL/2 (gFID **2.13** *vs*. 3.04) while still generating high-quality samples **74×** **faster**.

38th Conference on Neural Information Processing Systems (NeurIPS 2024).

# 1 Introduction

In recent years, image generation has experienced remarkable progress, driven by the significant advancements in both transformers [19, 65, 70, 10, 71, 72] and diffusion models [16, 58, 29, 52, 21]. Mirroring the trends in generative language models [51, 62], the architecture of many contemporary image generation models incorporate a standard image tokenizer and de-tokenizer. This array of models utilizes tokenized image representations—ranging from continuous [35] to discrete vectors [57, 64, 19]—to perform a critical function: translating raw pixels into a latent space. The latent space (*e.g.*, $32 \times 32$) is significantly more compact than the original image space ($256 \times 256 \times 3$). It offers a compressed yet expressive representation, and thus not only facilitates efficient training and inference of generative models but also paves the way to scale up the model size.

Although image tokenizers achieve great success in image generation workflows, they encounter a fundamental limitation tied to their intrinsic design. These tokenizers are based on an assumption that the latent space should retain a 2D structure, to maintain a direct mapping for locations between the latent tokens and image patches. For example, the top-left latent token directly corresponds to the top-left image patch. This restricts the tokenizer's ability to effectively leverage the redundancy inherent in images to cultivate a more compressed latent space.

Taking one step back, we raise the question *"is 2D structure necessary for image tokenization?"* To answer the question, we draw inspiration from several image understanding tasks where model predictions are based solely on high-level information extracted from input images —such as in image classification [17], object detection [8, 81], segmentation [67, 34, 74, 75], and multi-modal large language models [1, 41, 11]. These tasks do not need de-tokenizers, since the outputs typically manifest in specific structures other than images. In other words, they often format a higher-level 1D sequence as output that can still capture all task-relevant information. Prior arts, such as object queries [8, 67] or the perceiver resampler [1], encode images into a 1D sequence of a predetermined number of tokens (*e.g.*, 64). These tokens facilitate the generation of outputs like bounding boxes or captions. The success of these methods motivates us to investigate a more compact 1D sequence as image latent representation in the context of image reconstruction and generation. It is noteworthy that the synthesis of both high-level and low-level information is crucial for the generation of high-quality images, providing a challenge for extremely compact latent representations.

In this work, we introduce a transformer-based framework [65, 17] designed to tokenize an image to a 1D discrete sequence, which can later be decoded back to the image space via a de-tokenizer. Specifically, we present **T**ransformer-based 1-**Di**mensional **Tok**enizer (TiTok), consisting of a Vision Transformer (ViT) encoder, a ViT decoder, and a vector quantizer following the typical Vector-Quantized (VQ) model designs [19]. In the tokenization phase, the image is split and flattened into a series of patches, followed by concatenation with a 1D sequence of latent tokens. After the feature encoding process of ViT encoder, these latent tokens build the latent representation of the image. Subsequent to the vector quantization step [64, 19], the ViT decoder reconstructs the input images from the masked token sequence [15, 24].

Building upon TiTok, we conduct extensive experiments to probe the dynamics of 1D image tokenization. Our investigation studies the interplay between latent space size, model size, reconstruction fidelity, and generative quality. From this exploration, several compelling insights emerge:

1. Increasing the number of latent tokens representing an image consistently improves the reconstruction performance, yet the benefit becomes marginal after 128 tokens. Intriguingly, 32 tokens are sufficient for a reasonable image reconstruction.

2. Scaling up the tokenizer model size significantly improves performance of both reconstruction and generation, especially when number of tokens is limited (*e.g.*, 32 or 64), showcasing a promising pathway towards a compact image representation at latent space.

3. 1D tokenization breaks the grid constraints in prior 2D image tokenizers, which not only enables each latent token to reconstruct regions beyond a fixed image grid and leads to a more flexible tokenizer design, but also learns more high-level and semantic-rich image information, especially at a compact latent space.

4. 1D tokenization exhibits superior performance in generative training, with not only a significant speed-up for both training and inference but also a competitive FID score compared to a typical 2D tokenizer, while using much fewer tokens.

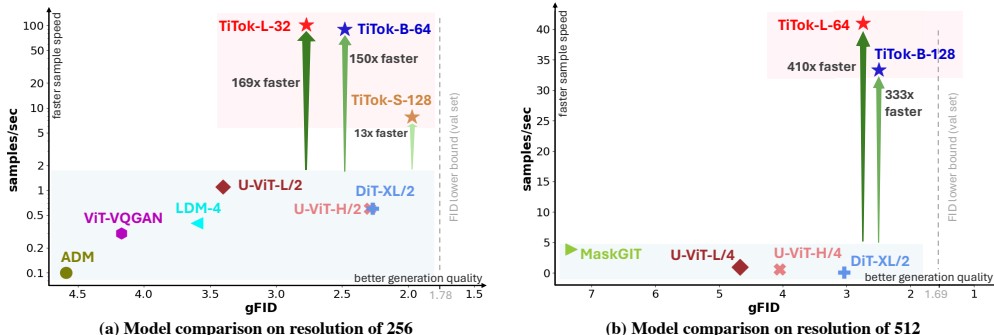

Figure 2: **A speed and quality comparison of TiTok and prior arts on ImageNet** $256 \times 256$ **and** $512 \times 512$ **generation benchmarks.** Speed-up is compared against DiT-XL/2 [52]. The sampling speed (de-tokenization included) is measured with an A100 GPU.

In light of these findings, we introduce the TiTok family, encompassing models of varying model sizes and latent sizes, capable of achieving highly compact tokenization with as few as **32 tokens**. We further confirm the model's efficacy in image generation through the MaskGIT [9] framework. TiTok is demonstrated to facilitate state-of-the-art performance in image generation, while requiring latent spaces that are $8\times$ to $64\times$ smaller, resulting in significant accelerations during both the training and inference phases. It also generates images with similar or higher quality but up to $410\times$ faster than state-of-the-art diffusion models such as DiT [18] (Fig. 2).

## 2 Related Work

**Image Tokenization.** Images have been compressed since the early days of deep learning with autoencoders [27, 66]. The general design of using an encoder that compresses high-dimensional images into a low-dimensional latent representation and then using a decoder to reverse the process, has proven to be successful over the years. Variational Autoencoders (VAEs) [35] extend the paradigm by learning to map the input to a distribution. Instead of modeling a continuous distribution, VQ-VAEs [50, 56] learn a discrete representation forming a categorical distribution. VQGAN [19] further improves the training process by using adversarial training [23]. The transformer design of the autoencoder is further explored in ViT-VQGAN [69] and Efficient-VQGAN [7]. Orthogonal to this, RQ-VAE [37] and MoVQ [80] study the effect of using multiple vector quantization steps per latent embedding, while MAGVIT-v2 [72] and FSQ [48] propose a lookup-free quantization. However, *all* aforementioned works share the same workflow of an image always being patchwise encoded into a *2D* grid latent representation. In this work, we explore an innovative *1D* sequence latent representation for image reconstruction and generation.

**Tokenization for Image Understanding.** For image understanding tasks (*e.g.*, image classification [17], object detection [8, 81, 78], segmentation [67, 74, 76], and Multi-modal Large Language Models (MLLMs) [1, 41, 77]), it is common to use a general feature encoder instead of an autoencoder to tokenize the image. Specifically, many MLLMs [41, 43, 61, 32, 22, 11] uses a CLIP [54] encoder to tokenize the image into highly semantic tokens, which proves effective for image captioning [13] and VQA [2]. Some MLLMs also explore discrete tokens [32, 22] or "de-tokenize" the CLIP embeddings back to images through diffusion models [61, 32, 22]. However, due to the nature of CLIP models that focus on high-level information, these methods can only reconstruct an image with high-level semantic similarities (*i.e.*, the layouts and details are not well-reconstructed due to CLIP features). Therefore, our method is significantly different from theirs, since the proposed TiTok aims to reconstruct both the high-level and low-level details of an image, same as typical VQ-VAE tokenizers [35, 57, 19].

**Image Generation.** Image generation methods range from sampling the VAE [35], using GANs [23] to Diffusion Models [16, 58, 29, 52, 21, 44] and autoregressive models [63, 12, 50]. Prior studies that are most related to this work build on top of a learned VQ-VAE codebook to generate images. Autoregressive transformer [19, 69, 7, 37], similar to decoder-only language models, model each patch in a step-by-step fashion, thus requiring as many steps as token number, *e.g.*, 256 or 1024. Non-

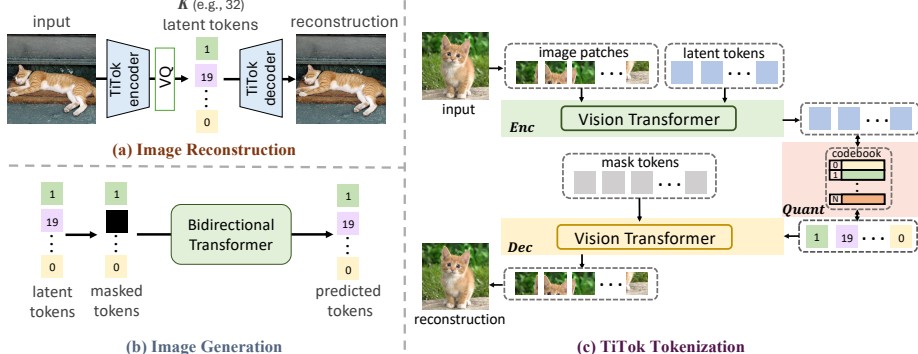

Figure 3: **Illustration of image reconstruction (a) and generation (b) with the TiTok framework (c).** TiTok contains an encoder $Enc$, a quantizer $Quant$, and a decoder $Dec$. Image patches, along with a few (*e.g.*, 32) latent tokens, are passed through the Vision Transformer (ViT) encoder. The latent tokens are then vector-quantized. The quantized tokens, along with the mask tokens [15, 24], are fed to the ViT decoder to reconstruct the image.

autoregressive (or bidirectional) transformers [80, 72, 68], such as MaskGIT [9], generally predict more than a single token per step and thus require significantly fewer steps to predict a complete image. Apart from that, further studies looked into improved sampling strategies [39, 40, 38]. As we focus on the tokenization stage, we apply the commonly used non-autoregressive sampling scheme of MaskGIT to generate a sequence of tokens that is later decoded into an image.

## 3 Method

### 3.1 Preliminary Background on VQ-VAE

The image tokenizer plays a pivotal role in facilitating the generative models by providing a compact image representation at latent space. For the scope of our discussion, we primarily focus on the Vector-Quantized (VQ) tokenizer [64, 19], given its broad applicability across various domains, including but not limited to image and video generation [19, 9, 58, 71], large-scale pretraining [12, 5, 49, 3, 18] and multi-modal models [20, 73].

A typical VQ model contains three key components: an encoder $Enc$, a vector quantizer $Quant$, and a decoder $Dec$. Given an input image $\mathbf{I} \in \mathbb{R}^{H \times W \times 3}$, where $H$ and $W$ denote the image's height and width, the image is initially processed by the encoder $Enc$ and converted to latent embeddings $\mathbf{Z}_{2D} = Enc(\mathbf{I})$, where $\mathbf{Z}_{2D} \in \mathbb{R}^{\frac{H}{f} \times \frac{W}{f} \times D}$, which downsamples the spatial dimensions by a factor of $f$. Subsequently, each embedding $z \in \mathbb{R}^D$ is mapped (via the vector quantizer $Quant$) to the nearest code $c_i \in \mathbb{R}^D$ in a learnable codebook $\mathbb{C} \in \mathbb{R}^{N \times D}$, comprising $N$ codes. Formally, we have:

$$Quant(z) = c_i, \text{ where } i = \underset{j \in \{1,2,...,N\}}{\operatorname{argmin}} \|z - c_j\|_2. \tag{1}$$

During de-tokenization, the reconstructed image $\hat{\mathbf{I}}$ is obtained via the decoder $Dec$ as follows:

$$\hat{\mathbf{I}} = Dec(Quant(\mathbf{Z}_{2D})). \tag{2}$$

Despite the numerous improvements over VQ-VAE [64] (*e.g.*, loss function [19], model architecture [69], and quantization/codebook strategies [80, 37, 72]), the fundamental workflow (*e.g.*, the 2D grid-based latent representations) has largely remained unchanged.

### 3.2 TiTok: From 2D to 1D Tokenization

While existing VQ models have demonstrated significant achievements, a notable limitation within the standard workflow exists: the latent representation $\mathbf{Z}_{2D}$ is often envisioned as a static 2D grid. Such a configuration inherently assumes a strict one-to-one mapping between the latent grids and the original image patches. This assumption limits the VQ model's ability to fully exploit the redundancies present in images, such as similarities among adjacent patches. Additionally, this approach constrains

the flexibility in selecting the latent size, with the most prevalent configurations being $f = 4$, $f = 8$, or $f = 16$ [58], resulting in 4096, 1024, or 256 tokens for an image of dimensions $256 \times 256 \times 3$. Inspired by the success of 1D sequence representations in addressing a broad spectrum of computer vision problems [8, 1, 41], we propose to use a 1D sequence, without the fixed correspondence between latent representation and image patches in 2D tokenization, as an efficient and effective latent representation for image reconstruction and generation.

**Image Reconstruction with TiTok.** To initiate our exploration, we establish a novel framework named **T**ransformer-based **1-D**imensional **Tok**enizer (TiTok), leveraging Vision Transformer (ViT) [17][1] to tokenize images into 1D latent tokens and subsequently reconstruct the original images from these 1D latents. As depicted in Fig. 3, TiTok employs a standard ViT for both the tokenization and de-tokenization processes (*i.e.*, both the encoder $Enc$ and decoder $Dec$ are ViTs). During tokenization, we patchify the image into patches (with a patch embedding layer) $\mathbf{P} \in \mathbb{R}^{\frac{H}{f} \times \frac{W}{f} \times D}$ (with patch size equal to the downsampling factor $f$ and embedding dimension $D$) and concatenate them with $K$ latent tokens $\mathbf{L} \in \mathbb{R}^{K \times D}$. They are then fed into the ViT encoder $Enc$. In the encoder output, we only retain the latent tokens as the image's latent representation, thereby enabling a more compact latent representation of 1D sequence $\mathbf{Z}_{1D}$ (with length $K$). This adjustment decouples the latent size from image's resolution and allows more flexibility in design choices. That is, we have:

$$\mathbf{Z}_{1D} = Enc(\mathbf{P} \oplus \mathbf{L}), \tag{3}$$

where $\oplus$ denotes concatenation, and we only retain the latent tokens from the encoder output.

In the de-tokenization phase, drawing inspiration from [15, 5, 24], we incorporate a sequence of mask tokens $\mathbf{M} \in \mathbb{R}^{\frac{H}{f} \times \frac{W}{f} \times D}$—obtained by replicating a single mask token $\frac{H}{f} \times \frac{W}{f}$ times—to the quantized latent tokens $\mathbf{Z}_{1D}$. The image is then reconstructed via the ViT decoder $Dec$ as follows:

$$\hat{\mathbf{I}} = Dec(Quant(\mathbf{Z}_{1D}) \oplus \mathbf{M}), \tag{4}$$

where the latent tokens $\mathbf{Z}_{1D}$ is first vector-quantized by $Quant$ and then concatenated with the mask tokens $\mathbf{M}$ before feeding to the decoder $Dec$.

Despite its simplicity, we emphasize that the concept of compact 1D image tokenization remains underexplored in existing literature. The proposed TiTok thus serves as a foundational platform for exploring the potentials of 1D tokenization and de-tokenization for natural images. It is worth noting that although one may flatten 2D grid latents into a 1D sequence, it significantly differs from the proposed 1D tokenizer, due to the fact that the implicit 2D grid mapping constraints still persist.

**Image Generation with TiTok.** Besides the image reconstruction task which the tokenizer is trained for, we also evaluate its effectiveness for image generation, following the typical pipeline [19, 9]. Specifically, we adopt MaskGIT [9] as our generation framework due to its simplicity and effectiveness, allowing us to train a MaskGIT model by simply replacing its VQGAN tokenizer with our TiTok. We do not make any other specific modifications to MaskGIT, but for completeness, we briefly describe its whole generation process with TiTok.

The image is pre-tokenized into 1D discrete tokens. At each training step, a random ratio of the latent tokens are replaced with mask tokens. Then, a bidirectional transformer takes the masked token sequence as input, and predicts the corresponding discrete token ID of those masked tokens. The inference process consists of multiple sampling steps, where at each step the transformer's prediction for masked tokens will be sampled based on the prediction confidence, which are then used to update the masked images. In this way, the image is "progressively generated" from a sequence full of mask tokens to an image with generated tokens, which can later be de-tokenized back into pixel spaces. The MaskGIT framework shows a significant speed-up in the generation process compared to auto-regressive models. We refer readers to [9] for more details.

### 3.3 Two-Stage Training of TiTok with Proxy Codes

**Existing Training Strategies for VQ Models.** Although most VQ models adhere to a straightforward formulation, their training process is notably sensitive, and the model's performance is heavily influenced by the adoption of more effective training paradigms. For instance, VQGAN [19] achieves

---

[1]Although other Transformer-based architectures (*e.g.*, Swin [45]) can also be used to instantiate TiTok, we adopt ViT for its simplicity and effectiveness.

a significant improvement in reconstruction FID (rFID) on the ImageNet [14] validation set, when compared to dVAE from DALL-E [55]. This enhancement is attributed to advancements in perceptual loss [33, 79] and adversarial loss [23]. Moreover, MaskGIT's modern implementation of VQGAN [9] utilizes refined training techniques without architectural improvements to boost the performance further. Notably, most of these improvements are exclusively applied during the training phase (*i.e.*, through auxiliary losses) and significantly affect the models' efficacy. Given the complexity of the loss functions, extensive tuning of hyper-parameters involved, and, most critically, the missing of a publicly available code-base for reference or reproduction [9, 69, 72], establishing an optimal experimental setup for the proposed TiTok presents a substantial challenge, especially when the target is a compact 1D tokenization which was rarely studied in literature.

**Two-Stage Training Comes to the Rescue.** Although training TiTok with the typical Taming-VQGAN [19] setting is feasible, we introduce a two-stage training paradigm for an improved performance. The two-stage training strategy contains "warm-up" and "decoder fine-tuning" stages. Specifically, in the first "warm-up" stage, instead of directly regressing the RGB values and employing a variety of loss functions (as in existing methods), we propose to train 1D VQ models with the discrete codes generated by an off-the-shelf MaskGIT-VQGAN model, which we refer to as *proxy codes*. This approach allows us to bypass the intricate loss functions and GAN architectures, thereby concentrating our efforts on optimizing the 1D tokenization settings. Importantly, this modification does not harm the functionality of the tokenizer and quantizer within TiTok, which can still fully function for image tokeniztion and de-tokenization; the main adaptation simply involves the processing of TiTok's de-tokenizer output. Specifically, this output, comprising a set of proxy codes, is subsequently fed into the same off-the-shelf VQGAN decoder to generate the final RGB outputs. It is noteworthy that the introduction of *proxy codes* differs from a simple distillation [26]. As verified in our experiments, TiTok yields significantly better generation performance than MaskGIT-VQGAN.

After the first training stage with proxy codes, we optionally have the second "decoder fine-tuning" stage, inspired by [10, 53], to improve the reconstruction quality. Specifically, we keep the encoder and quantizer frozen, and only train the decoder towards pixel space with the typical VQGAN training recipe [19]. We observe that such a two-stage training strategy significantly improves the training stability and reconstructed image quality, as shown in the experiments.

## 4 Experimental Results

### 4.1 Preliminary Experiments of 1D Tokenization

Building upon TiTok, we explore a range of configurations, including the model size and the number of tokens, to identify the most efficient and effective setup for a 1D image tokenizer. These preliminary experiments serve to provide a thorough evaluation, seeking a practical configuration of TiTok.

**Preliminary Experimental Setup.** Unless specified otherwise, we train all models with images of resolution $H = 256$ and $W = 256$, using the open-source MaskGIT-VQGAN [9] to supply proxy codes for training. The patch size for both tokenizer and de-tokenizer is established with $f = 16$, and the codebook $\mathbb{C}$ is configured to have $N = 1024$ entries with each entry a vector with 16 channels. For TiTok variants, we primarily investigate three model sizes—small, base, and large (*i.e.*, TiTok-S, TiTok-B, TiTok-L)—comprising $22M$, $86M$, and $307M$ parameters for encoder and decoder, respectively. We also assess the impact of varying the number of latent tokens $K$ from 16 to 256. We perform ablation experiments with an efficient setting (*e.g.*, shorter training).

**Evaluation Protocol.** Evaluation is conducted across multiple metrics to thoroughly assess the models, including both reconstruction and generation FID metrics (*i.e.*, rFID and gFID) [25] on the ImageNet dataset. We examine training/inference throughput to offer a direct comparison of generative model's efficiency relative to different latent sizes. Furthermore, given that the 1D VQ model inherently serves as a form of compact image compression, we further investigate the semantic information retained by the model through linear probing following MAE setting [24]. For the complete details of the training and testing protocols (*e.g.*, hyper-parameters, training costs), we refer the reader to the supplementary material Sec. A.

After the setup, we now summarize the preliminary experimental findings below.

**An Image Can be Represented by 32 Tokens.** The redundancy inherent in image representation is well-acknowledged, as evidenced by the practice of masking significant portions of images (*e.g.*,

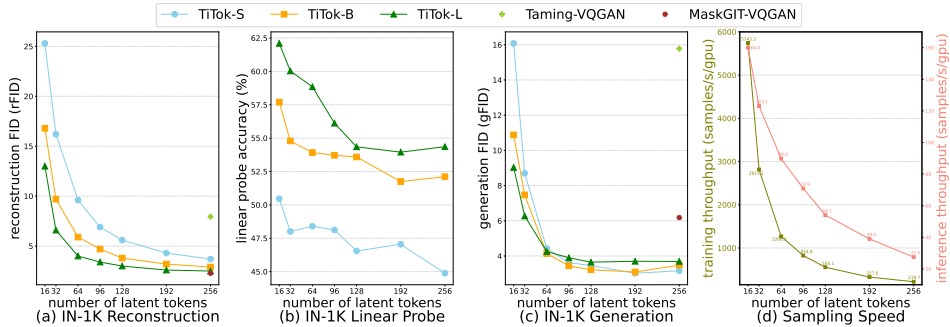

Figure 4: **Preliminary experimental results with different TiTok variants.** We provide a comprehensive exploration in (a) ImageNet-1K reconstruction. (b) ImageNet-1K linear probing. (c) ImageNet-1K generation. (d) Training and inference throughput of MaskGIT-ViT as generator and TiTok as tokenizer (evaluated on A100 GPUs, inference includes de-tokenization step with TiTok-B). Detailed numbers can be found in supplementary material Sec. B.

75% in MAE [24]) to expedite the training process without negatively affecting performance. This strategy has been validated across a variety of computer vision tasks that rely on high-level image features [30, 42]. However, the efficacy of such approaches in the context of image reconstruction and generation—where both low-level and high-level details are crucial for creating realistic reconstructed and generated outputs—remains underexplored. Consequently, in this experiment, we aim to determine the minimum number of tokens required to reconstruct and generate high-quality images. As depicted in Fig. 4a, although model performance progressively improves with an increase in the number of latent tokens, significant enhancements are predominantly observed when $K$ ranges from 16 to 128. Beyond this point, increasing the latent space size yields only marginal gains. *Intriguingly, we find that with merely* 32 *latent tokens, TiTok-L achieves performance better than a 2D VQGAN model [19] using* 256 *tokens.* This observation suggests that as few as 32 tokens may suffice as an effective image latent representation, optimizing the utilization of image redundancy.

**Scaling Up Tokenizer Enables More Compact Latent Size.** Another intriguing observation from Fig. 4a is that larger tokenizers facilitate more compact representations. Specifically, TiTok-B with 64 latent tokens achieves performance comparable to TiTok-S with 128 latent tokens, while TiTok-L with 32 latent tokens matches the performance of TiTok-B with 64 latent tokens. *This pattern indicates that with each incremental increase in TiTok size (*e.g.*, from S to B, or from B to L), it is possible to reduce the size of the latent image representation without compromising performance.* This trend underscores the potential benefits of scaling up the tokenizer to achieve even more compact image representations.

**Semantics Emerges with Compact Latent Space.** To evaluate the learned image representation, we perform linear probing experiments on the image tokenizer, as shown in Fig. 4b. Specifically, we add a batch normalization layer [31] followed by a linear layer on top of the frozen features from TiTok encoder, with all hyper-parameters strictly following the MAE protocol [24]. *We find that as the size of the latent representation decreases, the tokenizer increasingly learns semantically rich representations*, as indicated by the improved linear probing accuracy. This suggests that the model learns high-level information in scenarios of constrained representation space.

**Compact Latent Representation Improves Generative Training.** In addition to reconstruction capabilities, we assess TiTok's effectiveness and efficiency in generative downstream tasks, as illustrated in Fig. 4c and Fig. 4d. We note that variants of different tokenizer sizes yield comparable outcomes when the number of latent tokens is sufficiently large (*i.e.*, $K \geq 128$). However, within the domain of compact latent sizes (*i.e.*, $K \leq 64$), larger tokenizers notably enhance performance. Furthermore, the adaptability of 1D tokenization in TiTok facilitates more efficient and effective generative model training. For instance, model variants with $K = 32$, despite inferior reconstruction quality, demonstrate significantly better generative performance, *underscoring the advantages of employing a more condensed and semantically rich latent space for generative model training*. Additionally, the reduction in latent tokens markedly accelerates training and inference, with a $12.8\times$ increase in training speed (2815.2 *vs.* 219.7 samples/s/gpu) and a $4.5\times$ speed up sampling speed (123.1 *vs.* 27.5 samples/s/gpu), when utilizing $K = 32$ as opposed to $K = 256$.

Table 1: **ImageNet-1K** $256 \times 256$ **generation results evaluated with ADM [16].** †: Trained on OpenImages [36] ‡: Trained on OpenImages, LAION-Aesthetics/-Humans [59]. P: generator's parameters. S: sampling steps. T: throughput as samples per seconds on A100 with float32 precision.

| tokenizer | #tokens | codebook size | rFID↓ | generator | gFID↓ | P↓ | S↓ | T↑ |
|---|---|---|---|---|---|---|---|---|
| *diffusion-based generative models* | | | | | | | | |
| Taming-VQGAN† [58] | 1024 | 16384 | 1.14 | LDM-8 [58] | 7.76 | 258M | 200 | - |
| VAE† [58] | 4096×3 | - | 0.27 | LDM-4 [58] | 3.60 | 400M | 250 | 0.4 |
| | | | | UViT-L/2 [4] | 3.40 | 287M | 50 | 1.1 |
| VAE [60]‡ | 1024×4 | - | 0.62 | UViT-H/2 [4] | 2.29 | 501M | 50 | 0.6 |
| | | | | DiT-XL/2 [52] | 2.27 | 675M | 250 | 0.6 |
| *transformer-based generative models* | | | | | | | | |
| Taming-VQGAN [19] | 256 | 1024 | 7.94 | Taming-Transformer [19] | 15.78 | 1.4B | 256 | 7.5 |
| RQ-VAE [37] | 256 | 16384 | 3.20 | RQ-Transformer [37] | 8.71 / 7.55 | 1.4B / 3.8B | 64 | 16.1 / 9.7 |
| MaskGIT-VQGAN [9] | 256 | 1024 | 2.28 | MaskGIT-ViT [9] | 6.18 | **177M** | 8 | 50.5 |
| ViT-VQGAN [69] | 1024 | 8192 | 1.28 | VIM-Large [69] | 4.17 | 1.7B | 1024 | 0.3 |
| TiTok-L-32 | 32 | 4096 | 2.21 | MaskGIT-ViT [9] | 2.77 | **177M** | 8 | **101.6** |
| TiTok-B-64 | 64 | 4096 | 1.70 | MaskGIT-ViT [9] | 2.48 | **177M** | 8 | 89.8 |
| TiTok-S-128 | 128 | 4096 | 1.71 | MaskGIT-UViT-L [9, 4] | 2.50 / **1.97** | 287M | 8 / 64 | 53.3 / 7.8 |

## 4.2 Main Experiments

Based on the observations above, the proposed TiTok family effectively trades off a larger model size to a more compact latent size. In this section, we majorly focus on ImageNet generation benchmarks against prior arts, and evaluate TiTok as a tokenizer in the generative MaskGIT framework [9].

**Implementation Details.** We primarily investigate the following TiTok variants: TiTok-S-128 (*i.e.*, small model with 128 tokens), TiTok-B-64 (*i.e.*, base model with 64 tokens), and TiTok-L-32 (*i.e.*, large model with 32 tokens), where each variant designed to halve the latent space size while scaling up the model size. For resolution 512, we double the latent size to ensure more details are kept at higher resolution, leading to TiTok-L-64 and TiTok-B-128. In the final setting for TiTok training, the codebook is configured to $N = 4096$, and the training duration is extended to $1M$ iterations (200 epochs). We also adopt the "decoder fine-tuning" stage to further enhance model performance, where the encoder and quantizer are kept frozen and the decoder is fine-tuned for $500k$ iterations. For the training of generative models, we utilize the MaskGIT [9] framework without any specific modifications, except for the adoption of an arccos masking schedule [6]. All other parameters are the same as previous setups, and all design improvements will be verified in the ablation studies.

**Main Results.** We summarize the results on ImageNet-1K generation benchmark of resolution $256 \times 256$ and $512 \times 512$ in Tab. 1 and Tab. 2, respectively.[2]

For ImageNet $256 \times 256$ results in Tab. 1, TiTok can achieve a similar level of reconstruction FID (rFID) with a much smaller number of latent tokens than other VQ models. Specifically, using merely 32 tokens, TiTok-L-32 achieves a rFID of 2.21, comparable to the well trained VQGAN from MaskGIT [9] (rFID 2.28), while using $8\times$ smaller latent representation size. Furthermore, when using the same generator framework and same sampling steps, TiTok-L-32 improves over MaskGIT by a large margin (from 6.18 to 2.77 gFID), showcasing the benefits of a more effective generator training with compact 1D tokens. When compared to other diffusion-based generative models, TiTok can also achieve a competitive performance while enjoying an over $\mathbf{100}\times$ speed-up during the sampling process. Specifically, TiTok-L-32 achieves a better gFID than LDM-4 [58] (2.77 *vs.* 3.60), while generating images dramatically faster by **254** times (101.6 samples/s *vs.* 0.4 samples/s). Our best-performing variant TiTok-S-128 outperforms state-of-the-art diffusion method DiT-XL/2 [52] (gFID 1.97 *vs.* 2.27), with a $13\times$ speed-up.

For ImageNet $512 \times 512$ results in Tab. 2, the significantly better accuracy-cost trade-off of TiTok persists. TiTok maintains a reasonably good rFID compared to other methods, especially considering that TiTok uses much fewer tokens (*i.e.*, higher compression ratio). For generation, all TiTok variants

---

[2]For fairness, we mainly consider tokenizers with vanilla VQ modules. More advanced quantization methods [72, 48] may further benefit TiTok but beyond this paper's focus on 1D image tokenization. See supplementary material Sec. C for the complete table.

Table 2: **ImageNet-1K** $512 \times 512$ **generation results evaluated with ADM [16].** ‡: Trained on OpenImages, LAION-Aesthetics and LAION-Humans [59]. P: generator's parameters. S: sampling steps. T: throughput as samples per seconds on A100 with float32 precision.

| tokenizer | #tokens | codebook size | rFID↓ | generator | gFID↓ | P↓ | S↓ | T↑ |
|---|---|---|---|---|---|---|---|---|
| *diffusion-based generative models* | | | | | | | | |
| VAE [60]‡ | 4096×4 | - | 0.19 | UViT-L/4 [4] | 4.67 | 287M | 50 | 1.0 |
| | | | | UViT-H/4 [4] | 4.05 | 501M | 50 | 0.6 |
| | | | | DiT-XL/2 [52] | 3.04 | 675M | 250 | 0.1 |
| *transformer-based generative models* | | | | | | | | |
| MaskGIT-VQGAN [9] | 1024 | 1024 | 1.97 | MaskGIT-ViT [9] | 7.32 | **177M** | 12 | 3.9 |
| TiTok-L-64 | 64 | 4096 | 1.77 | MaskGIT-ViT [9] | 2.74 | **177M** | 8 | **41.0** |
| TiTok-B-128 | 128 | 4096 | 1.52 | MaskGIT-ViT [9] | 2.49 | **177M** | 8 | 33.3 |
| | | | | MaskGIT-ViT [9] | **2.13** | **177M** | 64 | 7.4 |

Table 3: **Ablation study improved final models for main experiments.** We ablate the tokenizer designs, and generator designs on ImageNet-1k benchmark. The final settings are labeled in gray. Generation results are based on tokenizers without decoder fine-tuning

(a) TiTok configuration. Results reported in accumulation manner

| TiTok-L-32 | rFID↓ | IS↑ |
|---|---|---|
| baseline | 6.59 | 110.3 |
| + larger codebook | 5.85 | 116.6 |
| + 200 epochs | 5.48 | 117.3 |
| + decoder finetuning | 2.21 | 195.5 |

(b) Masking schedules for generator with TiTok-L-32

| schedule | gFID↓ | IS↑ |
|---|---|---|
| cosine | 5.17 | 191.8 |
| arccos | 4.94 | 194.0 |
| linear | 4.95 | 193.7 |
| square root | 5.63 | 170.9 |

(c) Effects of *proxy codes*

| | rFID↓ | IS↑ |
|---|---|---|
| Taming-VQGAN training setting | | |
| Taming-VQGAN (2D) | 7.94 | - |
| TiTok-B-64 (2D) | 15.58 | 64.2 |
| TiTok-B-64 | 5.15 | 120.5 |
| Two-stage training with *proxy codes* | | |
| TiTok-B-64 | 1.70 | 195.2 |

significantly outperform our baseline MaskGIT [9] by a large margin. When compared with diffusion-based models, TiTok-L-64 shows a superior performance to DiT-XL/2 [52] (2.74 *vs.* 3.04), while running $410\times$ **faster**. The best-performing variant TiTok-B-128 can significantly outperform DiT-XL/2 by a large margin (2.13 *vs.* 3.04) but also generates high-quality samples $74\times$ **faster**. We also provide visualization results and analysis in supplementary material Sec. D.

## 4.3 Ablation Studies

We report the ablation studies regarding our final model designs in Tab. 3. Specifically, in Tab. 3a, we ablate the tokenizer designs on image reconstruction. We begin with our baseline TiTok-L-32 which attains 6.59 rFID. Employing a larger codebook size improves the rFID by 0.74, while further increasing the training iterations (from 100 epochs to 200 epochs) yields another 0.37 improvement of rFID. On top of that, the "decoder fine-tuning" (our stage-2 training strategy) can substantially improve the overall reconstruction performance to 2.21 rFID.

In Tab. 3b, we examine the effects of different masking schedules for MaskGIT with TiTok. Interestingly, unlike the original MaskGIT setting [9] which empirically found that the cosine masking schedule significantly outperforms the other schedules, we observe that MaskGIT equipped with TiTok changes the preference to the arccos or linear schedules. Additionally, unlike [9] which reported that the root schedule performs much worse than the others, we observe that TiTok is quite robust to different masking schedules. We attribute the observations to TiTok's ability to provide a more compact and more semantic meaningful tokens compared to 2D VQGAN, as compared to the cosine masking schedule, linear and arccos schedules have a lower masking ratio in the early steps. This coincides with the observation that masking ratio is usually higher for redundant signals (*e.g.*, 75% masking ratio in images [24]) while relatively lower for semantic meaningful inputs (*e.g.*, 15% masking ratio in languages [15]).

We ablate the effects of training paradigm in Tab. 3c. We begin with the training setting of Taming-VQGAN [19], where TiTok-B-64 obtains 5.15 rFID, outperforming the original 2D Taming-VQGAN's 7.94 rFID under the same training setting. We also show the necessity of 1D tokenization by building a 2D variant of TiTok-B64, where the architecture remains the same except that image patches instead of latent tokens are used as image representation. As a result, we observe that the 2D variant suffers from a much worse performance (15.58 *vs.* 5.15 rFID), since the fixed correspondences in 2D tokenization limited a reasonable reconstruction under compact latent space. This result

demonstrates the effectiveness of the proposed 1D tokenization, especially at a much more compact latent size. **Although TiTok can achieve a reasonably well performance under straightforward single-stage training, there exists a performance gap compared to the MaskGIT-VQGAN [9] due to the missing of a strong training recipe, of which no public reference or access exists.** Therefore, we adopt the two-stage training with *proxy codes*, which proves to be effective and can outperform the MaskGIT-VQGAN (1.70 *vs.* 2.28 rFID). It is noteworthy that the two-stage training is not that crucial to obtain a reasonable 1D tokenizer, and we believe that TiTok, with the simple single-stage Taming-VQGAN's training setting, could also benefit from training on a lagrer-scale dataset [36] as demonstrated in [58] and we leave it for future work due to the limited compute.

**Details on the Two-Stage Training.** We provide more technical details on the two-stage training. Specifically:

- In the first stage (warm-up stage), we use an off-the-shelf ImageNet-pretrained MaskGIT-VQ tokenizer to tokenize the input image into 256 tokens, which we refer to as proxy codes.

- In the first stage training, instead of regressing the original RGB values, we use the proxy codes as reconstruction targets. Specifically, the workflow is: RGB images are patchified and flattened into a sequence and concatenated with 32 latent tokens, then they are fed into TiTok-Enc (Encoder of TiTok). Later, the latent tokens are kept as token representation and go through the quantizer. The quantized latent tokens are concatenated with 256 mask tokens and go through the TiTok-Dec (Decoder of TiTok). And the final output mask tokens are supervised by proxy codes using cross-entropy loss.

- Afterwards, we freeze both the TiTok-Enc and quantizer, and then only fine-tune the TiTok-Dec (responsible for reconstructing proxy codes) and MaskGIT-Dec (responsible for reconstructing RGB values from proxy codes) end-to-end towards pixel space, where the training losses include L2 loss, perceptual loss, and GAN loss following the common VQGAN paradigm.

Moreover, we also note that two-stage training is not necessary for TiTok training, and it works fine with the commonly used and publicly available Taming-VQGAN recipe as is shown in Tab. 3c . In this case, the whole workflow is pretty straightforward, where the TiTok-Dec will instead directly reconstruct the images at pixel space.

However, the Taming-VQGAN recipe (developed more than 3 years ago) leads to an inferior FID score when compared to state-of-the-art tokenizers, putting TiTok at disadvantage when compared against other methods. Therefore we propose the two-stage training to benefit TiTok from the state-of-art MaskGIT-VQGAN tokenizer, which shares a similar architecture to Taming-VQGAN but has a significantly better score (rFID 2.28 v.s. 7.94).

We also note that TiTok can work well with single-stage recipe, and it is promising to incorporate the recent modern VQGAN recipe from [47, 68] in our preliminary experiments.

## 5 Conclusion

In this paper, we have explored a compact 1D tokenization TiTok for reconstructing and generating natural images. Unlike the existing 2D VQ models that consider the image latent space as a 2D grid, we provide a more compact formulation to tokenize an image into a 1D latent sequence. The proposed TiTok can represent an image with 8 to 64 times fewer tokens than the commonly used 2D tokenizers. Moreover, the compact 1D tokens not only significantly improve the generation model's training and inference throughput, but also achieve a competitive FID on the ImageNet benchmarks. We hope our research can shed some light in the direction towards more efficient image representation and generation models with 1D image tokenization.

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

## Appendix

In the supplementary materials, we provide the following additional details:

- The comprehensive training and testing hyper-parameters and training costs for TiTok (Sec. A).
- The detailed results of the preliminary experiments reported in main paper's Fig. 4(Sec. B).
- A more comprehensive comparison with more metrics and baselines (Sec. C).
- Qualitative visualizations (Sec. D).
- Limitation discussion (Sec. E).
- Broader Impacts discussion (Sec. F).
- Dataset Licenses (Sec. G).

## A  Training and Testing Protocols

For image reconstruction (**tokenizer**) **at preliminary experiments**, the training augmentation is confined to random cropping and flipping, following [19]. The training regimen spans a short schedule, featuring a batch size of $256$ over $500k$ training iterations, which correlates to roughly $100$ epochs on the ImageNet dataset. We employ the AdamW optimizer [46] with an initial learning rate of $1 \times 10^{-4}$ (with cosine decay) and weight decay $1 \times 10^{-4}$. We only adopt stage-1 training here (*i.e.*, only the "warm-up" training stage). For the **main experiments**, we adopt the improvements as shown in the ablation study (Tab. 3 in main paper), including longer training to 200 epochs and decoder fine-tuning, all other hyper-parameters remain the same. We use patch size 16 for all vision transformers at resolution $256 \times 256$ and increase it to 32 for resolution $512 \times 512$ to ensure a computation efficiency.

TiTok-L refers to using a ViT-L for TiTok encoder and decoder, and TiTok-B, TiTok-S refers to using ViT-B and ViT-S respectively. Moreover, the tokenizer training takes 64 A100-40G for 74 hours (TiTok-L-32), 32 A100-40G for 41 hours (TiTok-B-64), 32 A100-40G for 50 hours (TiTok-S-128), 32 A100-40G for 70 hours (TiTok-B-128 for resolution 512), and 64 A100-40G for 91 hours (TiTok-L-64 for resolution 512), respectively.

For image generation (**generator**) **at preliminary experiments**, we majorly build the training and testing protocols on top of [9]. Specifically, all images are pre-tokenized using center crop and random flipping augmentation, and then processed by MaskGIT [9] to generate images via the masked image modeling procedure. During inference, a cosine masking schedule is utilized with 8 steps. The generative models are trained with a batch size of $2048$ and $500k$ iterations to improve training efficiency. We use AdamW optimizer [46] with learning rate $2 \times 10^{-4}$ and weight decay $0.03$. The learning rate starts from $2 \times 10^{-4}$ and then decay to $1 \times 10^{-5}$ following a cosine decaying schedule. We apply a dropout probability of $0.1$ on the class condition. The only differences of **main experiments** are using an arccos masking schedule as discussed in the ablation study (main paper Tab. 3), all other hyper-parameters remain the same. We follow prior arts [19, 9] to generate $50k$ samples for generation FID evaluation. We also adopt classifier-free guidance [28] following prior arts [10, 72].

At ImageNet $256 \times 256$, we use guidance scale 4.5, temperature 9.5 for TiTok-L-32, guidance scale 3.0, temperature 11.0 for TiTok-B-64, guidance scale 2.0, temperature 3.0 for TiTok-S-128. At ImageNet $512 \times 512$, we use guidance scale 2.0, temperature 7.5 for TiTok-L-64, guidance scale 2.5, temperature 6.5 for TiTok-B-128.

The generator training takes 32 A100-40G for 12 hours (TiTok-L-32), 16 hours (TiTok-B-64), 29 hours (TiTok-S-128), 26 hours (TiTok-B-128 for resolution 512), 18 hours (TiTok-L-64 for resolution 512) respectively.

## B  Detailed Results of Preliminary Experiments

We summarize the detailed results for Fig. 4 of main paper in Tab. 4.

Table 4: **Detailed results of preliminary experiments in main paper.**

(a) reconstruction FID.

| #token | 16 | 32 | 64 | 96 | 128 | 192 | 256 |
|---|---|---|---|---|---|---|---|
| TiTok-S | 25.3 | 16.2 | 9.6 | 6.9 | 5.6 | 4.3 | 3.7 |
| TiTok-B | 16.8 | 9.7 | 5.9 | 4.7 | 3.8 | 3.2 | 2.9 |
| TiTok-L | 13.0 | 6.6 | 4.0 | 3.4 | 3.0 | 2.6 | 2.5 |

(b) generation FID.

| #token | 16 | 32 | 64 | 96 | 128 | 192 | 256 |
|---|---|---|---|---|---|---|---|
| TiTok-S | 16.1 | 8.7 | 4.4 | 3.6 | 3.5 | 3.0 | 3.2 |
| TiTok-B | 10.9 | 7.5 | 4.1 | 3.4 | 3.2 | 3.1 | 3.5 |
| TiTok-L | 9.0 | 6.3 | 4.3 | 3.9 | 3.7 | 3.7 | 3.7 |

(c) Linear Probing accuracy.

| #token | 16 | 32 | 64 | 96 | 128 | 192 | 256 |
|---|---|---|---|---|---|---|---|
| TiTok-S | 50.46 | 48.01 | 48.40 | 48.12 | 46.55 | 47.05 | 44.88 |
| TiTok-B | 57.70 | 54.79 | 53.92 | 53.72 | 53.59 | 51.75 | 52.11 |
| TiTok-L | 62.10 | 60.03 | 58.85 | 56.12 | 54.35 | 53.95 | 54.36 |

Table 5: **ImageNet-1K** $256 \times 256$ **generation results evaluated with ADM [16].** †: Trained on OpenImages [36] ‡: Trained on OpenImages, LAION-Aesthetics/-Humans [59]. P: generator's parameters. S: sampling steps. T: throughput as samples per seconds on A100 with float32 precision, measured with *w/ guidance* variants if available. "guidance" refers to classifier-free guidance.

| tokenizer | rFID↓ | generator | w/o guidance gFID↓ | w/o guidance IS↑ | w/ guidance gFID↓ | w/ guidance IS↑ | P↓ | S↓ | T↑ |
|---|---|---|---|---|---|---|---|---|---|
| *diffusion-based generative models* | | | | | | | | | |
| Taming-VQGAN† [58] | 1.14 | LDM-8 [58] | 15.82 | 78.82 | 7.76 | 209.5 | 258M | 200 | - |
| VAE† [58] | 0.27 | LDM-4 [58] | 10.56 | 103.5 | 3.60 | 247.7 | 400M | 250 | 0.4 |
| | | UViT-L/2 [4] | 9.03 | 111.5 | 3.40 | 219.9 | 287M | 50 | 1.1 |
| VAE [60]‡ | 0.62 | UViT-H/2 [4] | 6.60 | 142.5 | 2.29 | 263.9 | 501M | 50 | 0.6 |
| | | DiT-XL/2 [52] | 9.62 | 121.5 | 2.27 | 278.2 | 675M | 250 | 0.6 |
| *transformer-based generative models* | | | | | | | | | |
| Taming-VQGAN [19] | 7.94 | Taming-Transformer [19] | 15.78 | 78.3 | - | - | 1.4B | 256 | 7.5 |
| RQ-VAE [37] | 3.20 | RQ-Transformer [37] | 8.71 | 119.0 | - | - | 1.4B | 64 | 16.1 |
| | | | 7.55 | 134.0 | - | - | 3.8B | | 9.7 |
| MaskGIT-VQGAN [9] | 2.28 | MaskGIT-ViT [9] | 6.18 | 182.1 | - | - | **177M** | **8** | 50.5 |
| ViT-VQGAN [69] | 1.28 | VIM-Large [69] | 4.17 | 175.1 | - | - | 1.7B | 1024 | 0.3 |
| LFQ [72] | ∼0.9 | MAGVIT-v2 [72] | 3.65 | **200.5** | 1.78 | 319.4 | 307M | 64 | 1.1 |
| TiTok-L-32 | 2.21 | MaskGIT-ViT [9] | 3.15 | 173.0 | 2.77 | 199.8 | **177M** | **8** | **101.6** |
| TiTok-B-64 | 1.70 | MaskGIT-ViT [9] | **3.08** | **192.5** | 2.48 | 214.7 | **177M** | **8** | 89.8 |
| TiTok-S-128 | 1.71 | MaskGIT-UViT-L [9, 4] | 4.61 | 166.7 | 2.50 | 278.7 | 287M | 8 | 53.3 |
| | | | 4.44 | 168.2 | **1.97** | **281.8** | | 64 | 7.8 |

## C   Additional Results

We further report the class-conditional generation results comparison with more metrics and baselines in Tab. 5 and Tab. 6 for ImageNet $256 \times 256$ and $512 \times 512$ generation benchmarks, respectively. Moreover, we report both results without and with classifier-free guidance [28] under column "w/o guidance" and "w/ guidance" respectively.

As shown in Tab. 5, both TiTok-L-32 and TiTok-B-64 set a new state-of-the-art performance for results without classifier-free guidance (*i.e.*, w/o guidance column), while generating images at a much faster pace. Specifically, TiTok-L-32 achieves 3.15 gFID, surpassing current state-of-the-art MAGVIT-v2 [72]'s gFID 3.65, while requiring much fewer sampling steps (8 *vs.* 64) and smaller model size ($177M$ *vs.* $307M$), leading to a substantial sampling speed-up ($92.4\times$ faster, 101.6 *vs.* 1.1 samples/sec). Additionally, when compared to MaskGIT [9], which uses the exact same generator model (*i.e.*, MaskGIT-ViT) as ours and the only difference is the toeknizer, TiTok-L-32 achieves significantly a better performance (3.15 *vs.* 6.18). The improvement demonstrates the efficiency and effectiveness of the learned compact 1D latent space for image representation. When it comes to resolution $512 \times 512$ in Tab. 6, MaskGIT [9] requires 1024 tokens for image latent representation, while TiTok-L-64 requires $16\times$ fewer. As a result, when using the same generator (*i.e.*, MaskGIT-ViT), TiTok-L-64, w/o classifier-free guidance, not only significantly outperforms MaskGIT in terms of gFID (3.64 *vs.* 7.32) but also generates samples much faster. The advantages of TiTok become

Table 6: **ImageNet-1K** $512 \times 512$ **generation results evaluated with ADM [16].** †: Trained on OpenImages [36] ‡: Trained on OpenImages, LAION-Aesthetics/-Humans [59]. P: generator's parameters. S: sampling steps. T: throughput as samples per seconds on A100 with float32 precision, measured with *w/ guidance* variants if available. "guidance" refers to classifier-free guidance.

| tokenizer | rFID↓ | generator | w/o guidance gFID↓ | IS↑ | w/ guidance gFID↓ | IS↑ | P↓ | S↓ | T↑ |
|---|---|---|---|---|---|---|---|---|---|
| *diffusion-based generative models* | | | | | | | | | |
| VAE [60]‡ | 0.19 | UViT-L/4 [4] | 18.03 | 76.9 | 4.67 | 213.3 | 287M | 50 | 1.0 |
| | | UViT-H/4 [4] | 15.71 | 101.3 | 4.05 | 263.8 | 501M | 50 | 0.6 |
| | | DiT-XL/2 [52] | 12.03 | 105.3 | 3.04 | 240.8 | 675M | 250 | 0.1 |
| *transformer-based generative models* | | | | | | | | | |
| MaskGIT-VQGAN [9] | 1.97 | MaskGIT-ViT [9] | 7.32 | 156.0 | - | - | **177M** | 12 | 3.9 |
| LFQ [72] | 1.22 | MAGVIT-v2 [72] | 4.61 | 192.4 | - | - | 307M | 12 | 3.5 |
| | | | **3.07** | **213.1** | **1.91** | **324.3** | 307M | 64 | 1.0 |
| TiTok-L-64 | 1.78 | MaskGIT-ViT [9] | **3.64** | 179.8 | 2.74 | 221.1 | **177M** | **8** | **41.0** |
| TiTok-B-128 | 1.37 | MaskGIT-ViT [9] | 3.91 | **182.0** | 2.49 | 260.4 | **177M** | **8** | 33.3 |
| | | | 4.17 | 181.0 | **2.13** | **261.2** | | 64 | 7.4 |

even more significant when compared to the diffusion models such as DiT-XL/2 [52] with guidance: TiTok-L-64 not only shows a superior performance (2.74 *vs.* 3.04), but also enjoys a dramatically higher generation throughput ($410\times$).

An interesting observation is that under w/o guidance case, TiTok-L-32 (for 256 resolution) and TiTok-L-64 (for 512 resolution) can outperform most other methods, including TiTok-S-128 and TiTok-B-128, yet they benefit relatively less from the classifier-free guidance in the w/ guidance column. We note it indicates that the great potential of TiTok at compact latent size is still not fully unleashed yet, and better adaptation of inference time improvements for 1D compact tokens, such as classifier-free guidance, which was designed for methods with much more tokens and steps, could be a promising future direction.

## D   Visualizations

We provide visualization of the generated images using TiTok in Fig. 5 and Fig. 6. Moreover, we visualize the reconstruction results under different numbers of tokens and different model sizes in Fig. 7, where we observe that the model tends to keep the high-level layout or salient objects when the latent representation size is limited. Besides, a larger model size reconstructs an image with more details under a compact latent space size, demonstrating an effective way towards a more compact latent space.

We also provide more uncurated visualization samples in Fig. 8 and Fig. 9.

## E   Limitations

This paper proposes a novel 1D tokenization method designed to eliminate the fixed corresponding constraints of existing 2D tokenization methods. The 1D tokenization model is validated using the Vector Quantization (VQ) tokenizer formulation alongside a Masked Transformer generator framework. Despite the promising results, the proposed 1D tokenization formulation theoretically has the potential to generalize to other tokenizer formulations (*e.g.*, 1D-VAE), other generation frameworks (*e.g.*, Diffusion Models), and beyond the image modality (*e.g.*, video). However, exploring these extensions is beyond the scope of this paper due to limited computational resources, and we leave these as promising directions for future research.

## F   Broader Impacts

Generative models have numerous applications with diverse potential social impacts. While these models significantly enhance human creativity, they can also be misused for misinformation, harassment, and perpetuating social and cultural biases. Similar to other deep learning methods, generative

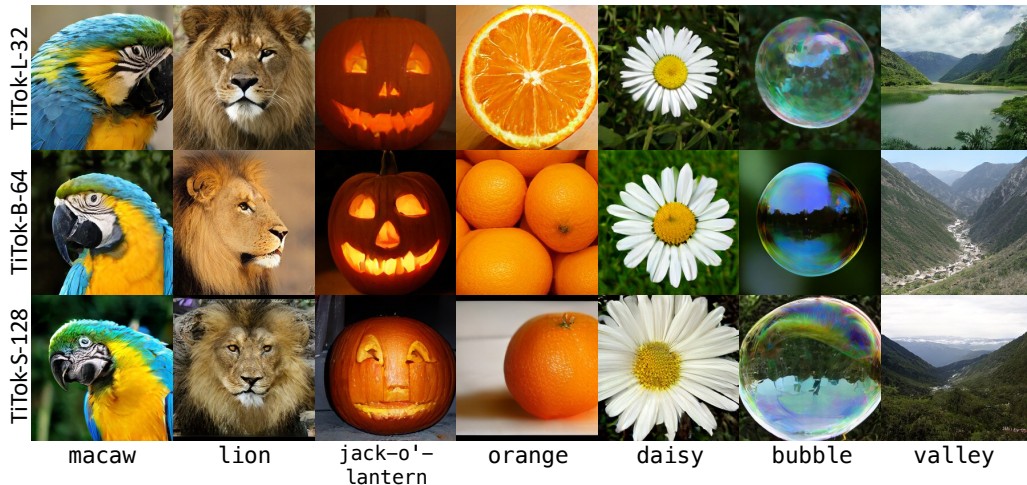

Figure 5: **Visualization of generated images from TiTok variants with MaskGIT [9].** Corresponding ImageNet class names are shown below the images.

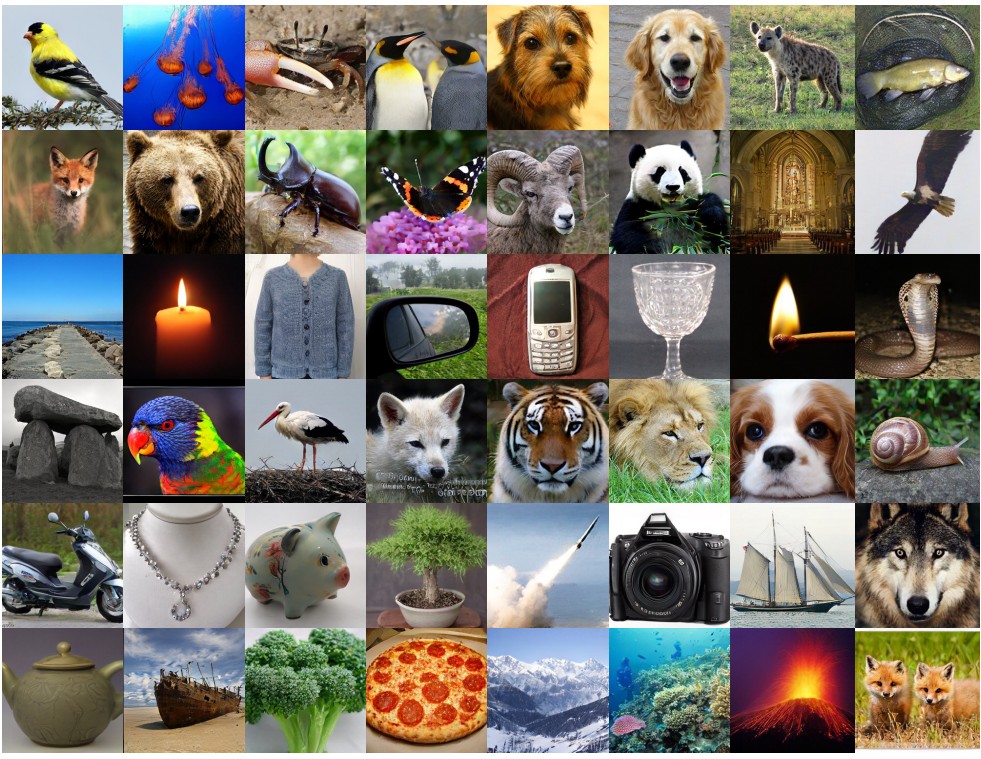

Figure 6: **Visualization of generated images from TiTok-L-32 with MaskGIT [9] across random ImageNet classes.**

models can be heavily influenced by dataset biases, leading to the reinforcement of negative social stereotypes and viewpoints. Developing unbiased models that ensure both robustness and fairness is a critical area of research. However, addressing these issues is beyond the scope of this paper.

Increasing number of latent token (K=16, 32, 64, 128, 256)

Increasing model size (S, B, L)

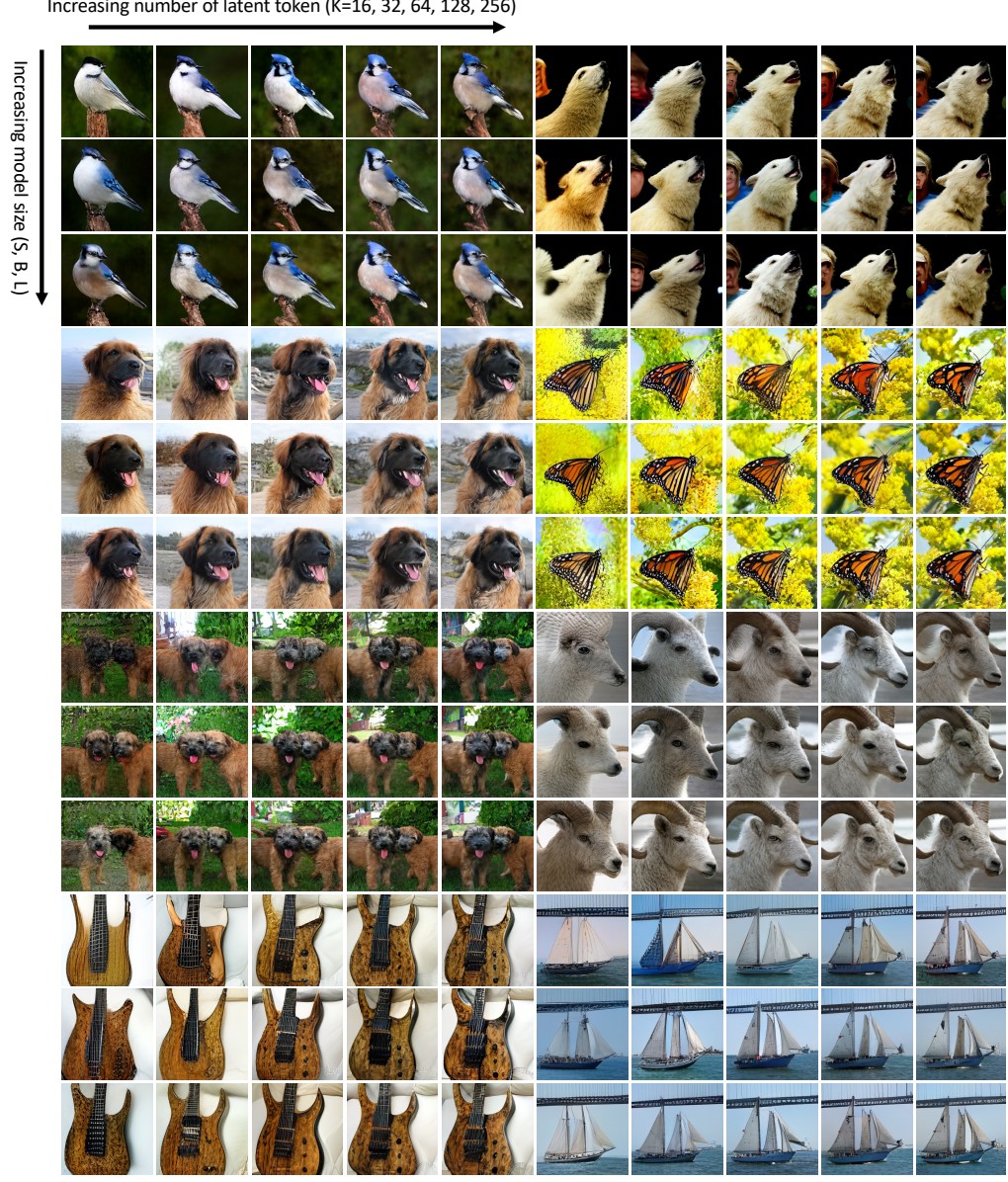

Figure 7: **Visual comparison of reconstruction results.** Scaling model size enables a better image quality while using a more compact latent space size. It is also observed that TiTok tends to keep the salient regions when latent space is limited.

Considering the potential risks, this paper is limited to class-conditional generation using a fixed, public, and controlled set of classes.

## G    Dataset Licenses

The datasets we used for training and/or testing TiTok are described as follows.

**ImageNet-1K:**    We train and evaluate TiTok on ImageNet-1K generation benchmark. This dataset spans 1000 object classes and contains 1,281,167 training images, 50,000 validation images and

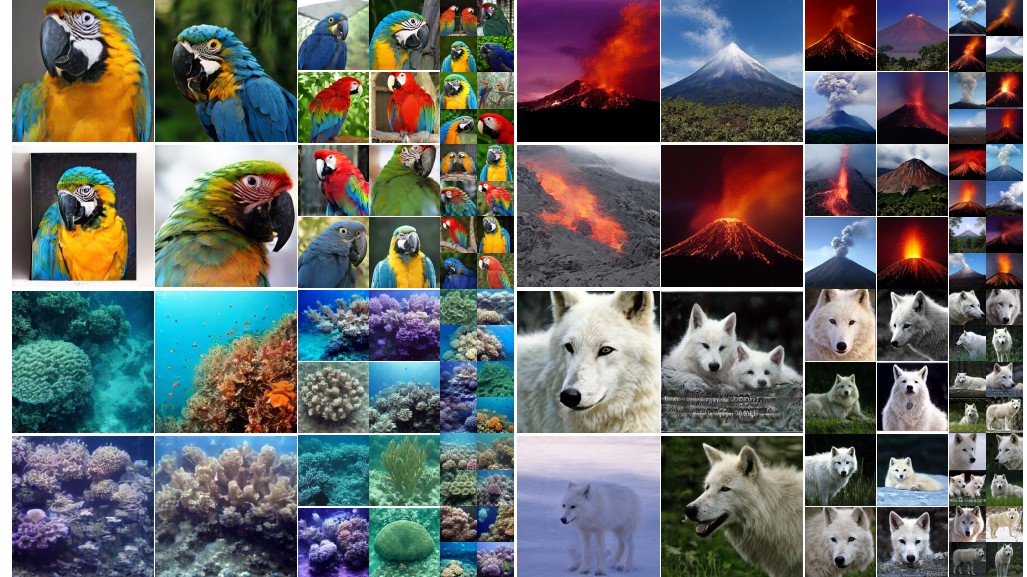

Figure 8: Uncurated $256 \times 256$ TiTok-L-32 samples. Class labels (class ids) from left to right and top to down are: "macaw" (88), "volcano" (980), "coral reef" (973), "white wolf" (270).

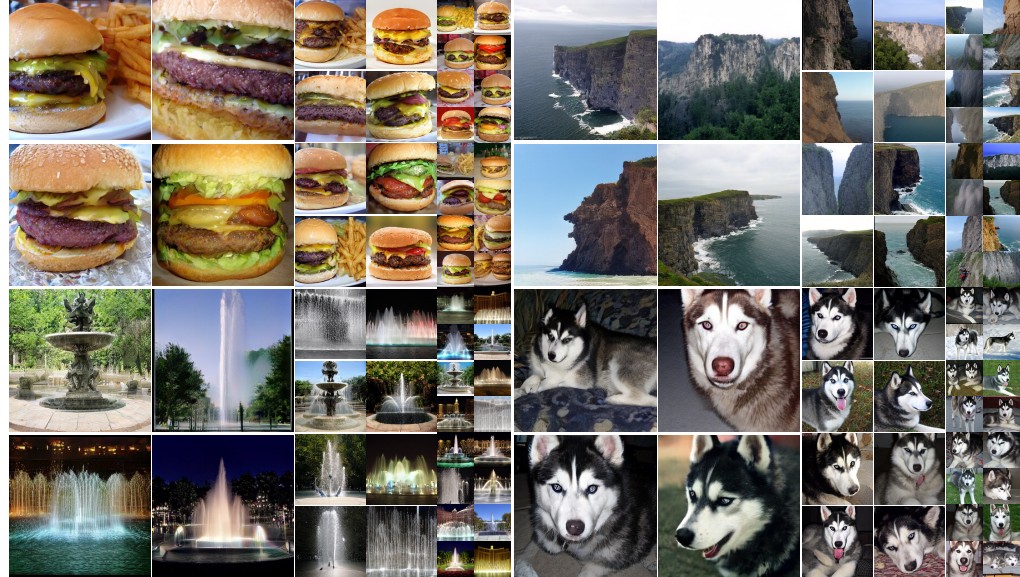

Figure 9: Uncurated $256 \times 256$ TiTok-L-32 samples. Class labels (class ids) from left to right and top to down are: "cheeseburger" (933), "cliff" (972), "fountain" (562), "Siberian husky" (250).

100,000 test images. We use the training set for our tokenizer and generator training. The validation set is used to compute reconstruction FID for evaluating tokenizers. The generation results are evaluated with generation FID using pre-computed statistics and scripts from ADM [16] [3].

License: https://image-net.org/accessagreement

URL: https://www.image-net.org/

---

[3]https://github.com/openai/guided-diffusion/tree/main/evaluations

