# OpenReview forum: "An Image is Worth 32 Tokens for Reconstruction and Generation"
_NeurIPS.cc/2024/Conference — NeurIPS 2024 poster_

### Official Review · Reviewer_HiP7 · 2024-06-26

**Soundness:** 4
**Presentation:** 4
**Contribution:** 4
**Rating:** 8
**Confidence:** 4

**Summary:**

This paper proposes a novel way to tokenize images to benefit image reconstruction and generation. This paper argues that the convention of compressing images into 2D latent spaces in the VQVAE/VQGAN setting limits the VQ model’s ability to fully exploit the redundancies present in images.

During encoding, a fixed number of latent tokens are concatenated with the image patches. A vision transformer (ViT) is used to extract the representation from image patches to the latent representation. During decoding, the latent tokens are concatenated with a set of masked tokens. Another ViT is used to reconstruct image patches from the latent representation. However, instead of training encoder and decoder end2end, this paper first trains the model with the discrete codes generated by an off-the-shelf MaskGIT-VQGAN model, and then only fine-tunes the decoder with RGB pixels. During image generation, MaskGIT is used to generate the latent tokens.

The model is tested on 256x256 ImageNet for image reconstruction, generation and classification.

**Strengths:**

1. The proposed method is simple yet effective.
2. Scaling up the tokenizer seems able to achieve even more compact image representations.
3. The experiments on image classification is very interesting. As the size of the latent representation decreases, the tokenizer increasingly learns semantically rich representations.

**Weaknesses:**

1. More efforts should be put in writing Two-Stage Training. It is not clear how exactly the "warmup" works. Does it train the whole encoder-decoder but with frozen codebook used in VQGAN? Or it means the model is trained by the quantized value of VQGAN instead of by the RGB pixels, i.e., the input and output of the model are the quantized value of VQGAN?
2. In the ablation test of 2D variant of TiTok-B64, how many tokens are used in this 2D representation? What if we use the same number of tokens for 1D and 2D variants? Will the performance be similar?

**Questions:**

How important is the off-the-shelf codebook? If the MaskGIT-VQGAN codebook is replaced by some other codebook, let's say FSQ, how would the performance look like? Can we use a continuous autoencoder instead of VQGAN as the outside compressor?

**Limitations:**

The limitation and potential negative societal impact have been discussed in the appendix.

---

> ### Author Rebuttal · Authors · 2024-08-07
>
> > **W1: More details on two-stage training?**
>
> Please see ***"General Questions and Concerns - Details of two-stage training."*** At the warm-up stage, the model's input is still RGB images, and output is the proxy codes, with cross-entropy loss for supervision.
>
> > **W2: Ablation test of 2D variant of TiTok-B64?**
>
> We thank the reviewer for the question. Both the TiTok-B-64 (2D) and TiTok-B-64 (1D) use the same number (64) of tokens, except that 2D variant kept the patch tokens as latent grid representation and directly reconstructed images from these 2D tokens (similar to a common 2D tokenizer), whereas 1D variant uses the latent tokens to guide the reconstruction from the mask token sequences. This experiment demonstrates that 1D formulation shows superior performance to 2D counterparts, especially when the number of tokens is limited.
>
> > **Q1: How important is the off-the-shelf codebook? Continuous autoencoder instead of VQGAN as the outside compressor?**
>
> Please see ***"General Questions and Concerns - Reviewer HiP7 Q1: How important is the off-the-shelf codebook? Can we use a continuous autoencoder?"***

---

> > ### Comment · Reviewer_HiP7 · 2024-08-10
> >
> > Thanks for the clarification. Since most reviewers have the confusion of the two-stage training, apparently this part is very poorly written. I strongly suggest the authors to revise the manuscript.

---

> > > ### Author Response · Authors · 2024-08-13
> > > **Thanks for Your Review and Support**
> > >
> > > Thank you so much for the valuable suggestions and for considering our responses! We will revise and refine the confusing parts in the two-stage training as discussed in the rebuttal as suggested. If you need any further information or clarification, please feel free to contact us!

---

### Official Review · Reviewer_Mn4Y · 2024-07-08

**Soundness:** 3
**Presentation:** 3
**Contribution:** 3
**Rating:** 6
**Confidence:** 4

**Summary:**

This paper introduces TiTok (Transformer-based 1-Dimensional Tokenizer) that can tokenize images as compact 1D sequences instead of 2D latent grids. Accompanied with bidirectional non-autoregressive image generator, TiTok achieves SOTA performance on imagenet 256x256 benchmark with much faster generation process. The key contributions are:
1) A 1D tokenization method that can represent images using significantly fewer tokens compared with traditional 2D approaches without comprising performance (as few as 32 tokens);
2) Scaling up the size of tokenizer model helps to learn more compact and semantic-rich latent representations;
3) Significant speed-up for both training and inference with competitive performance compared with 2D tokenizer.

**Strengths:**

1) The idea of using 1D representation instead of the conventional 2D is novel. It shows as few as 32 tokens are able to reconstruct an image well. The significant reduction of token numbers makes the training and inference much more efficient. Furthermore, scaling model sizes enables more compact representation, which provides insights into scaling laws for image tokenization.
2) The experiments are comprehensive, including different model sizes and token numbers. The ablation of different design choices are well designed. The evaluations on both image reconstruction and generation show superior performance compared with MaskGIT and VQGAN.
3) The paper is well rewritten, and easy to follow. The codes and models are publicly available.

**Weaknesses:**

A major concern is the two-stage training method. From Table 3 (c), the gap with and wo proxy codes is big, 1.7 vs 5.1 for rFID, and 195 vs 120 for IS. However, the proxy codes come from MaskGIT. The authors emphasized that due to the missing of a strong training recipe, the sing-stage method is lagging behind. Thus, it seems the training method is the most important one compared with architecture designs.

BTW, recently there is an open-source implementation for MAGVIT2 including training codes: https://github.com/TencentARC/Open-MAGVIT2
It will be great if the authors can adopt their training method and make the single-stage training work.

**Questions:**

Another related work is Seed, which also proposes a 1D tokenizer with semantic meaningful tokens (32 tokens). It's better to cite it:
Ge, Yuying, Yixiao Ge, Ziyun Zeng, Xintao Wang, and Ying Shan. "Planting a seed of vision in large language model." arXiv preprint arXiv:2307.08041 (2023).

**Limitations:**

No negative societal impact

---

> ### Author Rebuttal · Authors · 2024-08-07
>
> > **W1&2: two-stage training method, Improved public available tokenizer training recipe?**
>
> Please see ***"General Questions and Concerns - Reviewer Mn4Y W1&2: Two-stage training and better single-stage recipe from Open-MAGVIT2?"***
>
> > **Q1: Comparison to SEED?**
>
> We thank the reviewer for the suggestion. SEED is already cited and discussed in the Related Work L102 to L109. Specifically, BLIP, SEED, EMU, and other similar multi-modal LLMs build a tokenizer on top of CLIP encoder, which gives highly-semantic tokens. However, due to the nature of CLIP models that focus on high-level information, they can only feed the tokens into a diffusion model and only reconstruct an image with high-level semantic similarities, while the layouts and details may not be well reconstructed. (see Fig. 3 in the SEED paper as a reference).

---

> > ### Comment · Reviewer_Mn4Y · 2024-08-12
> >
> > Thanks for providing the detailed rebuttal, especially running the single-stage training experiment by borrowing the training recipe from open-magvit2. I will raise my score.

---

> > > ### Author Response · Authors · 2024-08-13
> > > **Thanks for Your Review and Support**
> > >
> > > Thank you very much for your insightful feedback and for considering our responses. We're glad we could address all your concerns satisfactorily. Your support in recommending our paper for acceptance is greatly appreciated. If you have any further questions, please feel free to let us know.

---

### Official Review · Reviewer_8XiP · 2024-07-12

**Soundness:** 3
**Presentation:** 3
**Contribution:** 3
**Rating:** 7
**Confidence:** 4

**Summary:**

This paper introduces a transformer-based image tokenizer (ViT) designed to convert 2D images into a 1D discrete sequence, named TiTok. The authors demonstrate that an image of size 256x256 can be discretized into a compact space of only 32 discrete tokens. This new tokenizer encodes semantic information, in contrast to other local 2D tokenizers. TiTok is trained using a two-stage process: initially, proxy codes are used to avoid the complexity of VQGAN loss, followed by a fine-tuning stage. Finally, they train a MaskGIT model to generate the discrete tokens, achieving an FID score of less than 2.

**Strengths:**

- The proposed method not only demonstrates increased speed compared to previous approaches but also showcases improved quality, as indicated by the FID score and the visualization.
- Thanks to the compact space, the tokenizer learns semantic features rather than local features, which suggests potential for interesting interpolation and image manipulation applications in the future.
- The method is original and presents a novel approach to addressing the generative task using discrete tokens in a 1D sequence.

**Weaknesses:**

- The method relies on proxy training for the tokenizer, but the motivation for this choice is unclear. The VQGAN loss (Reco, LPIPS, and GAN) is not overly complex to train, as evidenced by the availability of many open-source implementations. Utilizing proxy tokens is cumbersome since it requires a pre-trained network with the VQGAN loss.

- The paper may lack sufficient ablation studies on hyperparameters for sampling, such as the number of steps, the role of temperature, or the CFG.

- Providing uncurated visualizations of few samples from one or multiple classes could have helped readers better assess the diversity generated by the model.

**Questions:**

- Figure 4 shows better gFID compared to rFID when using only 32 tokens. How is this possible? How can you achieve a better FID than the actual reconstruction?

- Given that the number of tokens to predict is low (<128) and the structure is 1D, why not use a decoder-only transformer with next-token prediction? This approach would eliminate the burden of the sampling strategy and maintain a manageable number of forward passes thanks to the tokenizers

- I might have missed something, but how do you train TiTok with the proxy codes when the number of tokens output by the VQGAN are more than 32 and when the number of codes in the codebook are not the same?

**Limitations:**

/

---

> ### Author Rebuttal · Authors · 2024-08-07
>
> > **W1: Why proxy code (two-stage training) instead of VQGAN loss (recon + LPIPS + GAN)?**
>
> It is noteworthy that the two-stage training is ***not*** necessary, as is demonstrated in ***"General Questions and Concerns-Single-stage training for TiTok"***. We also show that TiTok can work well with the commonly used Taming-VQGAN recipe as shown in Tab.3c. However, as is discussed in L345 - 352, there is a performance gap between Taming-VQGAN recipe and MaskGIT-VQGAN recipe, where the latter one has a significantly better performance but provides no public reference or access. We adopt the two-stage training mainly for bridging the gap between the Taming-VQGAN recipe and other state-of-the-art tokenizers.
>
>
> > **W2: Ablation studies on hyperparameters for sampling**
>
> Following prior arts [10, 67], we did a grid search (with step 0.5) on the sampling hyper-parameters and reported the optimal ones. We add a further ablation study based on TiTok-L-32 as suggested below (the final setting is labeled in bold, each grid number is organized as IS/FID). The effects of using CFG or not is already provided in Tab. 6 in the submission ("w/o guidance" refers to no CFG and "w/ guidance" refers to CFG).
>
> | guidance_scale \ temperature | 8            | 8.5          | 9            | 9.5           | 10            | 10.5          | 11            |
> |------------------------------|--------------|--------------|--------------|---------------|---------------|---------------|---------------|
> | **3**                            | 197.9/2.78   | 188.8/2.77  | 178.0/2.77   | 169.9/2.84   | 161.6/2.94    | 156.1/3.10    | 150.2/3.27    |
> | **3.5**                          | 207.5/2.92   | 199.4/2.82   | 191.2/2.74   | 182.2/2.76    | 174.3/2.84    | 166.2/2.93    | 159.6/3.04    |
> | **4**                            | 217.9/3.02   | 209.8/2.89   | 200.6/2.80   | 192.7/2.77    | 184.1/2.75    | 176.1/2.80    | 170.5/2.89    |
> | **4.5**                          | 226.2/3.11   | 217.5/3.00   | 209.7/2.87   | **199.8/2.77**    | 193.7/2.77    | 184.7/2.78    | 177.8/2.80    |
> | **5**                            | 234.0/3.28  | 225.0/3.09  | 217.4/2.98   | 208.6/2.87    | 202.0/2.81    | 194.4/2.79    | 187.8/2.77    |
> | **5.5**                          | 241.9/3.42   | 233.0/3.23   | 222.4/3.04   | 215.7/2.96    | 208.7/2.88    | 200.7/2.83    | 194.7/2.79    |
> | **6**                            | 247.7/3.65   | 237.0/3.41   | 230.8/3.19   | 220.9/3.04    | 214.5/2.94    | 208.0/2.88    | 202.0/2.83    |
>
> > **W3: Uncurated visualizations**
>
> Thanks for the valuable comments, we provided uncurated visualizations in the attached PDF file.
>
> > **Q1: Why is gFID better than rFID?**
>
> We appreciate the good question. This is because rFID is computed against the real ImageNet val set, i.e., we compute the FID score between the reconstructed val set and the real ImageNet val set. However, gFID is computed against the virtual ImageNet reference statistics from OpenAI's ADM [D] (see "Sec 2.2 Sample Quality Metrics" in their paper). This is also why the gFID of the real val set is 1.78 instead of 0. The rFID and gFID are commonly correlated but they are not directly comparable. It is also noticeable that the evaluation protocols we follow are widely used by most prior works [4, 10, 35, 48, 54, 67].
>
> > **Q2: Why not use auto-regressive transformer for generation?**
>
> As discussed in the limitation section, we use MaskGIT generation framework as it is much more efficient compared to diffusion models or auto-regressive models, e.g., DiT-XL/2 is a 675M model, ViT-VQGAN is a 1.7B model, and the recent VAR requires training a 2.0B model for SOTA performance, while our model using MaskGIT only trains a 177M model and achieves better or comparable performance to aforementioned counterparts.
>
> > **Q3: "Token number mismatch" in the proxy code training?**
>
> Please see ***"General Questions and Concerns - Reviewer 8XiP Q3: How is TiTok trained with Proxy codes?"***
>
> [D] Dhariwal, et.al. "Diffusion models beat gans on image synthesis." NeurIPS 2021

---

### Official Review · Reviewer_Pxnc · 2024-07-15

**Soundness:** 1
**Presentation:** 3
**Contribution:** 2
**Rating:** 4
**Confidence:** 5

**Summary:**

Background: VQ-GAN with 2D grid of latent tokens and fixed downsampling factors.
This paper proposes to use 1D tokens instead of 2D tokens.

Key ideas:
* Redundancies: adjacent regions are similar. 2D grid explicitly couples the latents and the pixels in the same relative coordinates.
* 1D tokens are enough for discriminative tasks (classification, etc.). Maybe they are enough for generative tasks too.

Architecture:
* Transformer-based encoder (tokenizer) and decoder (de-tokenizer)

Training:
* Warm-up stage: train the 1D VQ model with the proxy codes (= discrete codes generated by an off-the-shelf MaskGIT-VQGAN model)
* Decoder fine-tuning stage: finetune the decoder with frozen encoder and quantizer

**Strengths:**

Technical advantages:
* More freedom in designing the architecture of 1D tokenizer
* More semantic-rich image embedding // -> definition of semantic-rich? grounding experiment?
* Improvement in FID
* Faster (70x ~ 410x) generation

Thorough analyses:
* Scaling experiment -> 32 tokens are sufficient for reconstruction.
* Larger tokenizer enables smaller number of latent tokens.
* Smaller number of latent tokens have clearer semantics than larger one, regarding linear probing.

**Weaknesses:**

W1. There is no analysis of 1D tokens. What are the advantages of 1D tokens that match the drawbacks of 2D tokens mentioned in the introduction section? What are the effects of masking a subset of 1D tokens compared to 2D tokens? The paper emphasizes the 1D latent sequence but does not analyze whether the observed characteristics (scaling, compact latent, etc.) are due to this 1D nature. A comparison with methods that further reduce the size of 2D tokens is needed.

W2. For reconstruction results, it is necessary to compare using MSE, PSNR, and SSIM. FID is not sufficient to evaluate “reconstruction.” Why do we need rFID instead of typical reconstruction metrics such as PSNR, SSIM, and LPIPS?

W3. The explanation of decoder fine-tuning is too sparse. Although not a novel contribution of this paper, it plays a significant role in performance improvement as shown in the ablation study, requiring more detailed explanation beyond just the “VQGAN training recipe.”


Misc.

Sentences could be easier to read.
- e.g., L60 ... the ViT decoder (is utilized to reconstruct -> reconstructs) the input images ...
- I recommend reading "Style, the basics of clarity and grace".

**Questions:**

Q1. What are the advantages of 1D compared to 2D, and what are the potential disadvantages? Please provide a brief explanation.

Q2. Is TiTok compatible with diffusion models? What is the reason for choosing MaskGIT over diffusion models or VAR (Scalable Image Generation via Next-Scale Prediction)? Appendix mentions computational burden. Is MaskGIT cheaper than the others?

Q3. Could you discuss the relationship between TiTok and "Vision transformers need registers"?
https://arxiv.org/abs/2309.16588

**Limitations:**

Yes in the appendix sections.

---

> ### Author Rebuttal · Authors · 2024-08-06
>
> > **W1.1: Grounding experiment and analysis of 1D tokens, advantages of 1D tokens against 2D tokens. What if we mask a subset?**
>
> Main advantages of 1D tokens are "semantic-meaningful" and "more compact". The "semantic-meaningful" is grounded by experiments in Fig. 4c, FID score in Tab. 1, 2, and visualization in Fig. 7, where we show that TiTok tends to learn more semantic-meaningful representation with a limited number of tokens. For "more compact", as evidenced by Fig.4d and Tab. 1, 2, TiTok uses much fewer tokens and leads to a substantial generation speed-up.
>
> Masking a subset could be challenging. As there is no concept of "pad token" or "placehold" in the learned codebook (regardless of 1D or 2D), it is not doable to "mask a subset" and then reconstruct an image. The best we can do is to randomly replace a subset of tokens with random tokens, yet it is still challenging to provide meaningful insights or comparison between 1D and 2D tokenizers.
>
> > **W1.2: Reducing sizes with 2D tokens**
>
> We emphasize that the scaling experiments aim at studying the model's behavior under different numbers of tokens, which is not necessarily related to 1D or 2D tokens. However, 1D token representation provides a flexible design to use arbitrary number of tokens, while 2D tokenizers often limit the token number choices among $k^2$ (where $k^2$ is one of [1024, 256, 64]), making it not suitable to study the scaling property with different number latent tokens. Besides, as shown in Tab. 3c, where the 2D variant of TiTok-B using 64 tokens ($8^2$) achieves significantly worse performance than the 1D variant of TiTok-B using the same number of tokens, indicating that 2D tokenizers are not good choices with limited number of tokens.
>
> > **W2: More evaluation metrics for reconstruction**
>
> FID and IS are the main metrics to evaluate the reconstruction in the context of tokenizer and generator [10, 35, 54, 64]. As suggested, we report additional metrics obtained using the same code-base to ensure fairness.
>
> |  | num_tokens | rFID↓ | IS↑    | PSNR↑  | SSIM↑   | MAE↓    | MSE↓    |
> |--------------|------------|------|-------|-------|--------|--------|--------|
> | MaskGIT-VQ      | 256        | 2.28 | 180.4 | 18.14 | 0.4386 | 0.0878 | 0.0188 |
> | TiTok-L-32   | 32         | 2.21 | **195.5** | 15.88 | 0.3635 | 0.1189 | 0.0300 |
> | TiTok-B-64   | 64         | **1.70** | 195.2 | 17.06 | 0.4023 | 0.1011 | 0.0234 |
> | TiTok-B-64 w/ improved training recipe | 64 | 2.43 | 179.3 | **19.01** | **0.4479** | **0.0782** | **0.0154** |
> | TiTok-S-128  | 128        | 1.71 | 177.3 | 17.73 | 0.4255 | 0.0929 | 0.0202 |
>
>
> The slightly worse PSNR/SSIM scores of TiTok variants match our observation that although TiTok can retain most important/salient information of the image, the high-frequency/details may be ignored/made up, as compensation for using fewer tokens. However, an improved training recipe (see ***“General Questions and Concerns - Single-stage training for TiTok”***) can significantly boost all these metrics.
>
> > **W3: More details on decoder-finetuning:**
>
> See ***“General Questions and Concerns - Reviewer Pxnc W3: Explanation of decoder fine-tuning?”***
>
>
> > **Q1: Potential advantages and disadvantages of 1D v.s. 2D**
>
> 1D tokenizer allows more flexible design and more semantic-meaningful and compact tokens compared to 2D, significantly speeding up the generation process while maintaining competitive scores. It is noteworthy that one may use the same large number of tokens in 1D tokenizer, thus making it a direct alternative to existing 2D tokenizers with similar or better performance.
>
> The main disadvantages are 1D tokenizers are far more under-explored, compared to 2D tokenizers, demanding for more research efforts to study its applications to different tasks (e.g., image editing, diffusion models, video generation, multi-modal LLM, etc.). Besides, the training paradigm of 1D tokenizer could be further refined, e.g., while we adopt the two-stage training in the paper, we currently observe promising results from an improved single-stage training recipe as suggested by Reviewer Mn4Y, which could be further improved.
>
>
> > **Q2: Other generation models (e.g., diffusion)?**
>
> As discussed in the limitation section, we use MaskGIT generation framework as it is much more efficient compared to diffusion models or auto-regressive models, e.g., DiT-XL/2 is a 675M model, ViT-VQGAN is a 1.7B model, and the recent VAR [D] requires a 2.0B model for attaining SOTA performance, while our model with MaskGIT only requires a 177M model and achieves better or comparable performance to aforementioned counterparts.
>
> The TiTok framework is totally compatible with the KL-VAE formulation that is widely used by diffusion models. As this paper mainly focuses on the tokenizer part, we believe the combination with diffusion models can be a promising future direction.
>
>
> > **Q3: Comparison to Vision transformers need registers?**
>
> Thanks for the suggestion. We will cite the paper (denoted as ViTreg below) for a discussion in a revision. TiTok uses latent tokens as the image representation, similar to ViTreg and other prior works. (a detailed discussion is available in L102-109). However, the two works have significant differences. ViTreg uses a set of latent tokens to allow a cleaner and more interpretable attention map, as the added latent tokens can help *alleviate the artifacts/outliers in the original self-attention map*. On the other hand, TiTok explicitly uses latent tokens to encode all the information needed to reconstruct the original image, where the latent tokens can be regarded as an information bottleneck between the tokenizer and de-tokenizer. The two works have distinctly different motivations and focuses.
>
> > **W4: Writing improvement**
>
> Thanks for the valuable suggestions and we will revise accordingly.
>
> [D] Tian, Keyu, et al. "Visual autoregressive modeling: Scalable image generation via next-scale prediction." arXiv:2404.02905 (2024).

---

> > ### Comment · Reviewer_Pxnc · 2024-08-14
> >
> > I thank the authors for the rebuttal. Sorry for coming in late.
> >
> > I am lowering my rating to BR because this paper is
> > * We did this and that. It results in this and that.
> > * *We do not know why. It works.*
> >
> > The core argument of this paper is using 1D tokens instead of 2D tokens. However, the paper does not provide the principle behind the advantage of 1D tokens over 2D tokens. 2D tokens are capable of representing anything which can be represented by 1D tokens because 2D tokens have more operational freedom. I value the analysis showing various behavior. But, the advantage of 1D tokens is supported by only the behavior, not the principle or theory.
> >
> > In short, I think using 1D tokens is worth publishing. But the content should be largely revised not to mislead the principle.
> >
> > (I am open to discussion during reviewer-ac discussion period.)

---

### Official Review · Reviewer_YNgD · 2024-07-21

**Soundness:** 3
**Presentation:** 3
**Contribution:** 3
**Rating:** 6
**Confidence:** 5

**Summary:**

This paper introduces TiTok, a 1D image tokenization method that can represent images using significantly fewer tokens compared to existing 2D approaches. The key contributions are:
1. A 1D tokenization scheme that breaks the fixed grid constraints of 2D tokenizers, allowing more flexible and compact image representations.
2. A dual-stage training strategy using proxy codes to improve tokenizer performance.
3. Extensive experiments demonstrating state-of-the-art performance on ImageNet generation benchmarks while using 8-64x fewer tokens and achieving 74-410x faster inference compared to leading diffusion models.

**Strengths:**

1. The conclusion is shocking. 32 tokens are enough to construct a picture. Although I know that the information in the picture is very redundant, this conclusion still shocks me.
2. The effect is still good.

**Weaknesses:**

1. Compared with 2D token, the main advantage of 1D image token (in my opinion) is not the efficiency and training data. Although 2D token is cumbersome, it supports the expansion of 2D dimension and has many other advantages. In applications, for higher resolution images and better effects, 2D token is still the mainstream and is difficult to be shaken by 1D token. The impact of 1D token on image generation may not be its main application. The most important value of 1D image token for me lies in its possibility of combining with multimodal technology. Due to its information compression ability and two-dimensional encoding method, 2D token is difficult to combine with language model to form a multimodal language model. And the encoder of the current multimodal language model has no decoder available. 1D token may be able to support multimodal language model and allow language model to directly output images. This is very potential. But this article does not have corresponding thinking, which is a pity.
2. Although the image formed by 32 tokens is shocking, there are still many questions to be answered. Compared with the case of more tokens, can the information encoded by each token be estimated? Can we know what specific information is forgotten in the process of token compression? Can 1D tokens encode 2D relationships? How do 1D tokens understand translation and rotation? Is 1D token similar to the concept of high-level dictionary? These important questions are not studied in depth in this article.
3. There are too few pictures shown in this article to make further judgments.

**Questions:**

I have raised many question in weaknesses.

**Limitations:**

yes

---

> ### Author Rebuttal · Authors · 2024-08-06
>
> > **W1.1: 2D v.s. 1D in image generation**
>
> We appreciate the valuable insights and comments. We note that the "2D advantages" of "higher resolution images and better effects" mainly stem from more tokens being used. While the 1D formulation in this paper mainly aims at a compact tokenization, we note that when using the similar number (perhaps slightly fewer) tokens, it is expected to work on par, if not better, to 2D counterparts.
>
> > **W1.2: 1D tokenization for multi-modal large language model**
>
> We appreciate the suggestion and we strongly agree that TiTok has great potential in the multi-modal large language model. However, due to limitations on computation resources (e.g., The VQ-tokenizer-based multi-modal large language model Chameleon [C] by Meta requires training on 1024 A100-80G for 1 month), we thus mainly verify the tokenizer's effectiveness on the generation tasks, following prior arts [19, 35, 64, 66, 67]. We leave the application to multi-modal models as promising future directions.
>
> > **W2.1: Measure the information encoded by tokens**
>
> We thank the reviewer for the valuable questions. We tried our best to provide quantitative and qualitative measurements on the information carried by 1D tokens. For example, in Fig.4.c, we show the linear probing accuracy (and thus correlates with the "semantic" in the tokens) when the number of tokens varies. We also provide visualization on different numbers of tokens in Fig. 7 where we observe that the tokens tend to carry important/salient regions when the number of tokens is limited. Although we agree that it will be intriguing if we may measure each token individually, it may require further research efforts and we leave it for future work.
>
> > **W2.2: What specific information is forgotten? 2D relationship, translation, rotation? High-level dictionary?**
>
> It is noteworthy that our work is a very simple and initial attempt at this promising direction. As mentioned in W2.1, it is very challenging to directly "visualize" what role each token is responsible for, or how 2D relationship is modeled in the tokenization, although they are interesting topics towards better explainability. Based on experiments from Fig.4 and visualization from Fig. 7, we observe that the model tends to forget the low-level, high-frequency details in the background. For a similar reason, though the tokenizer can handle certain levels of translation/rotation in the input images, it is tricky to figure out which tokens are responsible for this part. High-level dictionary may not be capable to faithfully reconstruct the original images while TiTok tokens aim at a better trade-off between compact and semantic tokenization and faithful reconstruction (Please also see our response to Reviewer mn4Y’s Q1 for detailed examples on reconstructing images from CLIP high-level features, which fail to reconstruct the image layout or details).
>
> > **W3: More figures?**
>
> We appreciate the suggestions. We provided more visualization in the attached PDF file. As promised in the paper, we will also open-source the code & model for the community to examine.
>
> [C] Team, Chameleon. "Chameleon: Mixed-modal early-fusion foundation models." arXiv preprint arXiv:2405.09818 (2024).

---

> > ### Comment · Reviewer_YNgD · 2024-08-07
> > **Post-rebuttal review**
> >
> > I have read the author response and the review from other reviewers. I will raise my score to "weak accept". Thanks.

---

> > > ### Author Response · Authors · 2024-08-09
> > > **Thanks for Your Review and Support**
> > >
> > > Thank you sincerely for your valuable feedback and for taking our responses into account. We're pleased that we were able to address your concerns. We greatly appreciate your support in recommending our paper for acceptance. Should you have any additional questions, please feel free to let us know.

---

### Author Rebuttal · Authors · 2024-08-06

# General Questions and Concerns

We thank all reviewers for the initial positive scores and acknowledgements. We address the shared concerns below and upload additional visualization in the PDF attachment.

> **Details of two-stage training:**

To begin with, we describe the two-stage training in detail as follows:

1. In the first stage (warm-up stage), we use an off-the-shelf ImageNet-pretrained MaskGIT-VQ tokenizer to tokenize the input image into 256 tokens, which we refer to as proxy codes.

2. In the first stage training, instead of regressing the original RGB values, we use the proxy codes as reconstruction targets. Specifically, the workflow is: RGB images are patchified and flattened into a sequence and concatenated with 32 latent tokens, then they are fed into TiTok-Enc (Encoder of TiTok). Later, the latent tokens are kept as token representation and go through the quantizer. The quantized latent tokens are concatenated with 256 mask tokens and go through the TiTok-Dec (Decoder of TiTok). And the final output mask tokens are ***supervised by proxy codes using cross-entropy loss***.

3. After 2. is finished, we freeze both the TiTok-Enc and quantizer, and then only fine-tune the TiTok-Dec (responsible for reconstructing proxy codes) and MaskGIT-Dec (responsible for reconstructing RGB values from proxy codes) end-to-end towards pixel space, where the training losses include L2 loss, perceptual loss, and GAN loss following the common VQGAN paradigm.

> **Single-stage training for TiTok:**

As shown in Tab. 3c and discussed in L345-352. Two-stage training is ***not*** necessary for TiTok training, and it works fine with the commonly used and publicly available Taming-VQGAN recipe. In this case, the whole workflow is pretty straightforward, where the TiTok-Dec will instead directly reconstruct the images at pixel space.

However, the Taming-VQGAN recipe (developed more than 3 years ago) leads to an inferior FID score when compared to state-of-the-art tokenizers, putting TiTok at disadvantage when compared against other methods. Therefore we propose the two-stage training to benefit TiTok from the state-of-art MaskGIT-VQGAN tokenizer, which shares a similar architecture to Taming-VQGAN but has a significantly better score (rFID 2.28 v.s. 7.94)

We appreciate the reference to Open-MAGVIT2 per Reviewer Mn4Y's suggestion and some other recently open-sourced tokenizer training code bases such as LlamaGen (note that both are made publicly available after this submission). With these improved training recipes, we obtain much better score under the single-stage training, as shown below:

| Model                                          | rFID | IS    |
|------------------------------------------------|------|-------|
| Taming-VQGAN (256 tokens)                      | 7.94 | -     |
| MaskGIT-VQ (256 tokens)                        | 2.28 | 180.4 |
| TiTok-B-64 (single-stage w/ Taming-VQGAN recipe)| 5.15 | 120.5 |
| TiTok-B-64 (two-stages recipe)                 | 1.70 | 195.2 |
| TiTok-B-64 (***single-stage w/ improved recipe***) | 2.43 | 179.3 |

While there still exists some minor gap to the two-stage recipe, we believe the gap would be bridged further with better hyper-parameter tuning, etc., which we do not have time or resources to fully explore during the short rebuttal period.


> **Reviewer Pxnc W3: Explanation of decoder fine-tuning?**

We appreciate the question and please see the detailed two-stage workflow above in ***Details of two-stage training***.


> **Reviewer 8XiP Q3: How is TiTok trained with Proxy codes?**

We appreciate the question and please see the detailed two-stage workflow above in ***Details of two-stage training***. As proxy codes are used as reconstruction targets and we use the mask token sequence to formulate the de-tokenization process, the number of proxy codes and number of latent tokens do not matter in this training stage.

> **Reviewer Mn4Y W1&2: Two-stage training and better single-stage recipe from Open-MAGVIT2?**

We appreciate the feedback and reference to Open-MAGVIT2. Please see the table above  in ***Single-stage training for TiTok*** for single-stage TiTok with the updated training recipe.

> **Reviewer HiP7 Q1: How important is the off-the-shelf codebook? Can we use a continuous autoencoder?**

We thank the reviewer for the valuable questions. The role of off-the-shelf tokenizer is important for state-of-the-art performance but not for a usable tokenizer as demonstrated in ***Single-stage training for TiTok***. To the best of our knowledge, MaskGIT-VQ offers both a strong performance tokenizer and an open-source checkpoint (no training recipe though), which is why we use MaskGIT-VQ as the off-the-shelf tokenizer for two-stage training. Unfortunately, FSQ has not released their weight, making it challenging for us to experiment with it. In the table above in ***Single-stage training for TiTok***, we demonstrate that the off-the-shelf tokenizer can be removed while maintaining a similar performance, if we have a similar training recipe used by state-of-the-art tokenizers. We hope it can address the concerns regarding the off-the-shelf codebook.

Regarding using an off-the-shelf autoencoder such as KL-VAE, we believe it is totally feasible. However, regressing the continuous embedding requires more careful designs (e.g., GMM prediction head [A] or diffusion head [B]), while using VQ models simplifies the problems into classifying proxy codes. Thus we believe this is an interesting problem to be explored in the near future.

[A] Tschannen, Michael, et al. "Givt: Generative infinite-vocabulary transformers." arXiv preprint arXiv:2312.02116 (2023).

[B] Li, Tianhong, et al. "Autoregressive Image Generation without Vector Quantization." arXiv preprint arXiv:2406.11838 (2024).

---

### Decision · Program_Chairs · 2024-09-25

**Decision:**

Accept (poster)

**Comment:**

The paper introduces TiTok, a 1D image tokenizer that reduces token count while preserving image quality. Reviewer YNgD acknowledges the ability to reconstruct images with just 32 tokens but notes the paper doesn't explore potential in multimodal applications and leaves questions about how 1D tokens capture 2D relationships unanswered. Reviewer Pxnc raises concerns about insufficient analysis of 1D tokens' advantages over 2D tokens and the lack of standard reconstruction metrics like PSNR and SSIM. Reviewer 8XiP likes the method's originality and efficiency but points out the reliance on proxy training and suggests more ablation studies. Reviewer Mn4Y values the novelty but worries that the two-stage training with proxy codes might be the key contributor rather than the architecture itself. Reviewer HiP7 finds the method simple and effective but calls for clearer explanations of the two-stage training process.

The rebuttal clarifies the two-stage training process, explaining how they use an off-the-shelf tokenizer to generate proxy codes in the first stage and fine-tune the decoder in the second. They demonstrate that single-stage training with improved recipes achieves similar performance, addressing concerns about reliance on proxy codes.

All the reviewers vote to accept the paper except Pxnc who complains the paper lacks principles. The AC feels that the novelty of the paper should interest a broad audience of NeurIPS community and warrants publication.